# MxB is an interferon-induced restriction factor of human herpesviruses

Michel Crameri[1,2], Michael Bauer [2,3], Nicole Caduff[2,4], Raphael Walker[1], Fiona Steiner[1,2], Francesca D. Franzoso[5,6], Cornelia Gujer[4], Karin Boucke[3], Talissa Kucera[1], Andrea Zbinden[1], Christian Münz[4], Cornel Fraefel[5], Urs F. Greber [3] & Jovan Pavlovic[1]

The type I interferon (IFN) system plays an important role in controlling herpesvirus infections, but it is unclear which IFN-mediated effectors interfere with herpesvirus replication. Here we report that human myxovirus resistance protein B (MxB, also designated Mx2) is a potent human herpesvirus restriction factor in the context of IFN. We demonstrate that ectopic MxB expression restricts a range of herpesviruses from the *Alphaherpesvirinae* and *Gammaherpesvirinae*, including herpes simplex virus 1 and 2 (HSV-1 and HSV-2), and Kaposi's sarcoma-associated herpesvirus (KSHV). MxB restriction of HSV-1 and HSV-2 requires GTPase function, in contrast to restriction of lentiviruses. MxB inhibits the delivery of incoming HSV-1 DNA to the nucleus and the appearance of empty capsids, but not the capsid delivery to the cytoplasm or tegument dissociation from the capsid. Our study identifies MxB as a potent pan-herpesvirus restriction factor which blocks the uncoating of viral DNA from the incoming viral capsid.

[1] Institute of Medical Virology, University of Zurich, Winterthurerstrasse 190, 8057 Zürich, Switzerland. [2] Life Science Zurich Graduate School, Winterthurerstrasse 190, 8057 Zürich, Switzerland. [3] Institute of Molecular Life Sciences, University of Zurich, Winterthurerstrasse 190, 8057 Zürich, Switzerland. [4] Institute of Experimental Immunology,  University of Zurich, Winterthurerstrasse 190, 8057 Zürich, Switzerland. [5] Institute of Virology, University of Zurich, Winterthurerstrasse 266a, 8057 Zürich, Switzerland. [6] Department of Neurosurgery and Neuropathology, University Hospital Zurich, Frauenklinikstrasse 10, 8091 Zürich, Switzerland. These authors contributed equally: Cornel Fraefel, Urs F. Greber.  Correspondence and requests for materials should be addressed to J.P. (email: pavlovic.jovan@virology.uzh.ch)

Viruses are ubiquitous and affect all forms of life. In humans, livestock and plants, they cause disease by recurrent or chronic infections, and have had a major impact on human evolution. For example, viruses influenced the emergence and maintenance of elaborate host defence systems, such as innate and adaptive immunity[1–4]. In the course of an infection, viruses elicit danger signals, and trigger cell-based innate immunity owing to viral components, including virions, viral proteins and nucleic acids, and cell damage[5,6]. A major branch of mammalian innate immunity against viruses is the interferon (IFN) system[7]. This system comprises type I, II and III IFN, and involves hundreds of IFN-stimulated genes (ISGs). ISG expression can lead to the so-called restriction factors which directly inhibit a process essential to the production of virus progeny, or ISG expression can act indirectly by building up an elaborate antiviral network[8].

Human myxovirus resistance protein B (MxB) has recently been identified as an IFN-induced restriction factor of human immunodeficiency virus type 1 (HIV-1) and other primate lentiviruses[9–11]. MxB is a member of the dynamin-like large GTPases and is closely related to MxA, an ISG product with broad antiviral activity against a multitude of RNA viruses and some DNA viruses but not against herpes simplex virus 1 (HSV-1) or HIV-1[9,12,13]. At the subcellular level, MxB can be found in the cytoplasm, on the cytoplasmic face of nuclear membranes and in the nucleus[10,12,14]. Under certain conditions, MxB has been shown to bind to HIV-1 capsid-like recombinant assemblies via its amino-terminal domain[15–17]. This has been taken as evidence that MxB prevents uncoating of proviral DNA from incoming HIV-1 capsids. The antivirally active form of MxB is likely an antiparallel dimer and the formation of higher-order oligomeric structures does not appear to be required to exert full antiviral function[15,18]. However, binding of MxB to HIV-1 capsids is clearly not sufficient for the observed antiviral activity, since MxB retains binding capacity to capsid mutants that are resistant to the antiviral effect of MxB[15]. Intriguingly, the GTP-hydrolysing function of MxB appears to be dispensable for the antiviral activity against HIV-1[9,10,19], which stands in contrast to the antiviral activity of MxA that requires GTPase function[13,20]. Antiviral activity of MxB is likely not restricted to HIV-1, as indicated by a recent analysis of *MX2* (the gene encoding MxB) evolution[21]. In their study, Mitchell et al.[21] found that amino acid residues, which are important for anti-lentivirus activity, have not evolved under diversifying selection in primates, indicating that MxB functions beyond lentivirus restriction. We therefore evaluated whether other human viruses, which replicate in the nucleus, are restricted by MxB and focused on HSV-1 and human adenovirus C serotype 5 (HAdV-C5).

HAdV-C5 is a widespread human pathogen of the respiratory tracts and is life-threatening in immunosuppressed individuals[22]. Its replication is suppressed by IFN and can lead to persistent infection with low levels of virus production[23]. HAdV-C5 enters epithelial cells by clathrin-dependent and clathrin-independent, dynamin-2-dependent endocytosis[24]. It escapes from non-acidified early endosomes, is transported by dynein-dependent and microtubule-dependent transport to the nucleus, binds and uncoats at the nuclear pore complex (NPC) and imports a double-stranded linear DNA genome in complex with viral proteins into the nucleus[25–30]. Adenovirus intercepts IFN restriction by its immediate-early protein E1A, which inhibits the E3 ubiquitin ligase hBre1, and thereby results in transcriptional suppression of ISGs[31,32].

Human herpesviruses are prevalent in humans, and cause disease ranging from subclinical manifestations to encephalitis and cancer, particularly in immunocompromised individuals.

Members of each subfamily *Alphaherpesvirinae*, *Betaherpesvirinae* and *Gammaherpesvirinae* establish lifelong persistence by latent infections. The virions contain a double-stranded DNA genome wrapped in an icosahedral capsid surrounded by a proteinaceous layer referred to as tegument, and a lipid envelope that harbours the glycoproteins required for entry into host cells. Entry can either occur through fusion of the viral envelope with the plasma membrane or by endocytosis[33,34]. Upon entry, the tegument gradually dissociates and the capsid is transported along microtubules to the nuclear envelope. Injection of the viral genomic DNA into the nucleus occurs at the NPC in a process that involves tegument and capsid proteins as well as cellular nuclear import factors and nucleoporins (reviewed in Refs. [35–38]). Nuclear entry is followed by transcription of immediate-early genes including *RL2*, encoding the infected cell protein 0 (ICP0), which in turn drives HSV-1 early and late gene expression and is required for efficient replication.

Incoming herpesviruses trigger a pronounced innate immune response including the production of type I IFN[39,40]. Although the recognition of herpesviruses by pattern recognition receptors mediating IFN synthesis has been studied extensively (reviewed in Refs. [41–43]), little is known about the IFN-induced effector mechanisms that result in the inhibition of virus replication. Several ISGs with antiviral function such as the dsRNA-dependent protein kinase R (PKR), viperin, tetherin, zinc-finger antiviral protein and 2′–5′ oligoadenylate synthetase (OAS) have been implicated in restricting herpesviruses, but their contribution remains to be elucidated (reviewed in Refs. [44,45]).

Here we report that MxB efficiently blocks herpesvirus infection at an early post-entry step, which precludes the nuclear translocation of the incoming viral DNA.

## Results

**Interferon-induced MxB inhibits HSV-1 replication**. To assess the impact of MxB expression on HSV-1 infection, we transfected T98G glioblastoma cells known to express high levels of MxB in response to type I IFN[46] (see also Supplementary Fig. 1a) with non-targeting control small interfering RNA (siRNA) (siNT) and *MX2*-targeting siRNAs (siMxB 3'-untranslated region (UTR), siMxB #1). MxB was expressed in IFN-α2-treated cells transfected with control siRNA, where it accumulated at the nuclear envelope as well as in the cytoplasm, as previously described[10,47]. Transfection with *MX2*-specific siRNAs prevented the accumulation of MxB to detectable levels (Fig. 1a–c). Infection of siNT-transfected T98G cells with two distinct HSV-1 strains (MacIntyre and F) yielded viral titres of approximately $5 \times 10^7$ TCID$_{50}$/ml. In order to exclude non-specific effects of siRNA transfection, relative titres are shown (Fig. 1b, c). Pretreatment of T98G cells with 500 IU/ml IFN-α2 reduced viral titres of both strains approximately 1000-fold. Remarkably, in T98G cells transfected with siRNAs specific for *MX2*, the observed IFN-mediated inhibition of HSV-1 infection was partially released, resulting in approximately 10-fold increased viral titres (Fig. 1b, c). Increased titres in cells transfected with *MX2*-specific siRNAs were reflected in markedly enhanced levels of HSV-1 virion protein 16 (VP16) when compared to cells transfected with control siRNA (Fig. 1b, c).

We next monitored the influence of MxB expression on HSV-1 growth in an infection kinetics experiment by employing a live-cell fluorescence imaging system and a recombinant HSV-1 strain encoding green fluorescent protein (GFP, strain C12[48]). We infected T98G cells with C12 virus following transfection with non-targeting or *MX2*-specific siRNAs and stimulation with IFN-α2. In the absence of IFN-α2 stimulation, the C12 strain grew rapidly, reaching maximal GFP signal (40–56 arbitrary units (a. u.)) between 60 and 72 h post infection (p.i.), irrespective of the

siRNA used (Fig. 1d and Supplementary Fig. 1b). By contrast, in IFN-α2-stimulated cells transfected with non-targeting siRNA, HSV-1 grew slowly, reaching approximately 6 a.u. at 72 h p.i. In cells pretreated with *MX2*-specific siRNAs, however, the block

was gradually released and GFP signals reached between 27% and 45% of the IFN-α2-untreated controls (Fig. 1d). Hence, MxB contributes significantly to the IFN-mediated inhibition of HSV-1 infection. As a control, we performed an analogous

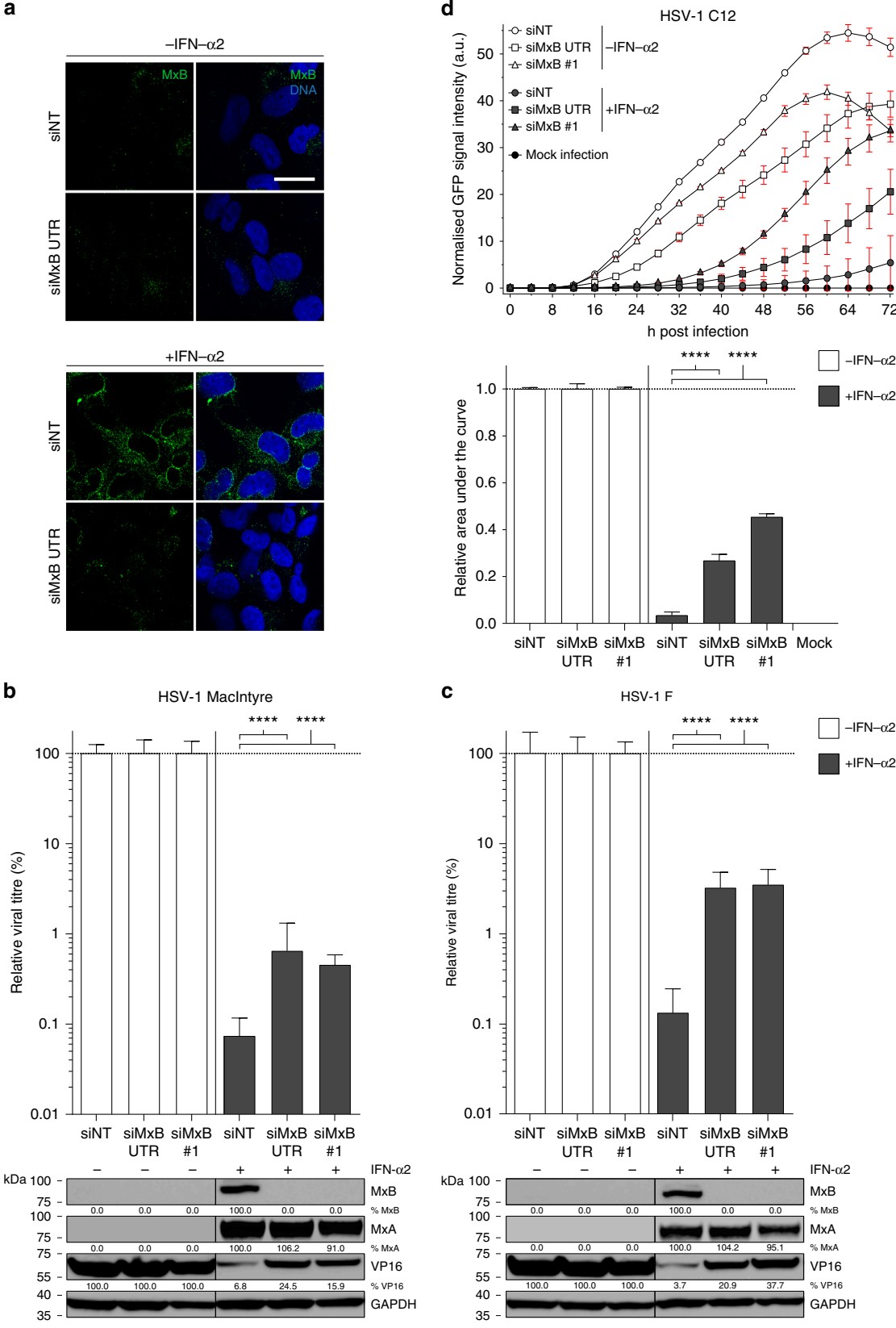

experiment using two siRNAs targeting the coding region of *PKR*, an ISG with known anti-HSV-1 activity[44]. The observed *PKR* siRNA-dependent release of HSV-1 restriction resulted in 22–26% of viral growth compared to the untreated controls (Supplementary Figs. 1c, d and 7d). We therefore concluded that in this setting, the potential of MxB to restrict HSV-1 is comparable to PKR.

**Overexpression of MxB inhibits herpesvirus growth**. We next tested whether ectopic expression of MxB in the absence of IFN-α2 stimulation would restrict HSV-1 and other members of the *Herpesviridae* family. For this purpose, we generated several stably transduced A549 and Vero cell clones constitutively expressing glutathione *S*-transferase M1 (GST, control) or MxB by means of lentiviral vectors. In MxB-expressing A549 cells, ectopic MxB was found at the nuclear membrane and in a punctate pattern in the cytoplasm (Fig. 2a). A similar phenotype was observed in Vero cells stably overexpressing MxB (Supplementary Fig. 2c). Transfection with siRNAs against the coding region of *MX2* (siMxB #1, siMxB #2) reduced the level of MxB expression to about 30% or less of the control (siNT), whereas an siRNA against the 3′-UTR of the endogenous *MX2* mRNA (siMxB UTR) not present in the overexpressed gene had no inhibitory effect (Fig. 2b and Supplementary Fig. 2d). Control infection of A549-GST and A549-MxB cell clones with the HIV-1-based luciferase reporter virus NL-Luc[49] and influenza A virus (IAV) confirmed that MxB restricts replication of HIV-1 but not IAV[9–12] (Supplementary Fig. 5b and Fig. 2c). Vesicular stomatitis virus (VSV) was previously reported to be inhibited by MxB[50]. In our cell culture system, we observed a fivefold reduction of VSV infection in cells expressing MxB as compared to GST (Fig. 2d). Further, we tested whether MxB would inhibit growth of human HAdV-C5. We observed a 2.4-fold reduction of HAdV-C5 titres in multi-round infections of A549-MxB cells (Fig. 2e), and an inhibition of GFP expression in single-round infections using GFP or late viral protein expression as a readout (Supplementary Fig. 5c). However, transfection of siRNA against *MX2* did not restore HAdV-C5 reporter gene expression, suggesting that the effects of MxB on HAdV-C5 infection were indirect or unspecific (Supplementary Fig. 5d). In contrast, we found that MxB expression had a pronounced negative effect on several representatives of the *Alphaherpesvirinae* subfamily. In our experiments, we used stocks of three different HSV-1 strains (MacIntyre, F or C12) grown under identical conditions. When we challenged A549 cells expressing MxB with these three strains, viral titres were reduced about 75-fold as compared to A549-GST

cells (Fig. 2f–h). Similarly, the commonly used HSV-2 strain G showed a reduction of approximately 50-fold (Fig. 2i). Transfection of an siRNA against the 3′-UTR of endogenous *MX2* mRNA had no effect on HSV growth. On the contrary, transfection with siRNAs against the coding region of *MX2* largely restored titres of all tested HSV-1 strains and HSV-2 (Fig. 2f–i). Cell lysates prepared at the time of virus supernatant collection revealed that HSV late protein VP16 was expressed at considerably lower levels in A549-MxB cells compared to control cells, yet VP16 expression was fully restored upon *MX2* silencing (Supplementary Fig. 7b). Importantly, siRNA transfection did not impact herpesvirus titres or VP16 expression in A549-GST cells, and the effect of the individual siRNAs on cell viability in the different cell lines was negligible, thereby excluding unspecific effects of the different siRNAs used in this system (Supplementary Fig. 7a). In order to rule out possible effects of individual cell clones on HSV titres, two additional, independently generated A549-MxB cell clones (MxB_2 and MxB_3) were tested for restriction of HSV-1 strain C12 and HSV-2. Both additional A549-MxB clones inhibited HSV-1 and HSV-2 to the same extent as the original A549-MxB clone, and this effect was again reverted with *MX2*-specific siRNAs (Supplementary Fig. 2a, b). The fact that the rescue of the HSV titres was not always complete (Fig. 2f–i, Supplementary Fig. 2b) may be due to incomplete silencing of *MX2* expression in A549-MxB cells at the time of infection (Fig. 2b and Supplementary Fig. 7e). In Vero cells, the MxB-mediated inhibition of HSV-1 replication was readily measurable but less pronounced as compared to A549 cells (Supplementary Fig. 2e). We then tested whether Kaposi's sarcoma-associated herpesvirus (KSHV), a representative of the *Gammaherpesvirinae* subfamily, was restricted by MxB in A549 cells. Here, we made use of a recombinant KSHV that expresses GFP upon virus entry into the host cell nucleus and red fluorescent protein (RFP) upon lytic reactivation[51]. While KSHV was detected in 79–88% of A549 cells expressing GST, only 27–31% of A549 cells expressing MxB showed a detectable level of KSHV infection (Fig. 3a). Pretreatment of A549-MxB cells with *MX2*-specific siRNAs siMxB #1 and siMxB #2 resulted in a strong rescue of KSHV infectivity (58–62% in A549-MxB cells compared to 70–87% in A549-GST cells, Fig. 3a). Moreover, we employed immunofluorescence assays to assess the accumulation of latency-associated nuclear antigen (LANA), one of the three main KSHV proteins expressed during persistent latent infection. Endogenous LANA staining is widely used to detect KSHV infection, as KSHV establishes latency as a default program in various systems in vitro and in vivo, including epithelial cells[51–54]. In A549-GST

**Fig. 1** Interferon-induced MxB inhibits HSV-1 replication. **a** T98G human glioblastoma cells were transfected with non-targeting siRNA (siNT) or siRNA targeting endogenous *MX2* (siMxB UTR). At 30 h post transfection, cells were mock-stimulated or stimulated with human IFN-α2 (1000 IU/ml) for 18 h. MxB protein expression and intracellular localisation was assessed by immunostaining. Nuclei were stained with Hoechst 33342. Scale bar, 20 μm. **b**, **c** T98G cells were transfected with non-targeting siRNA (siNT) or two different siRNAs targeting endogenous *MX2* (siMxB UTR, siMxB #1). At 30 h post transfection, cells were mock-stimulated or stimulated with human IFN-α2 (500 IU/ml). At 48 h post transfection, cells were infected with HSV-1 strain MacIntyre (**b**) or F (**c**) at a multiplicity of infection (MOI) of 0.1 for 32 h. Cell culture supernatants were subjected to TCID$_{50}$ assay and cells were lysed and pooled for immunoblot analysis of MxB protein expression, *MX2* silencing efficiency and HSV-1 late protein VP16 expression. MxA served as a control for siRNA specificity and GAPDH served as a loading control. Titres of mock-stimulated samples are each set to 100%. Bars indicate relative mean ± s.d., $n = 6$ biological replicates. ANOVAs modelling log-transformed relative titre with IFN and siRNA as explanatory variables: HSV-1 MacIntyre: IFN, $F_{(1,30)} = 1215.74$, $p < 0.0001$; siRNA, $F_{(2,30)} = 13.67$, $p < 0.0001$; antagonising interaction, $F_{(2,30)} = 14.81$, $p < 0.0001$. HSV-1 F: IFN, $F_{(1,30)} = 587.29$, $p < 0.0001$; siRNA, $F_{(2,30)} = 39.55$, $p < 0.0001$; antagonising interaction, $F_{(2,30)} = 33.63$, $p < 0.0001$. **d** T98G cells were siRNA-transfected and stimulated with human IFN-α2 as in **b**, but then mock-infected or infected with HSV-1 recombinant strain C12 expressing GFP (MOI = 0.1). Virus growth was determined by real-time quantification of GFP fluorescence. Upper panel: Data points indicate the integrated green object intensity normalised to cell confluence (mean ± s.d., $n = 3$ biological replicates). Lower panel: Relative area under the curve from 0 to 72 h post infection. GFP signals of mock-stimulated samples are each set to 1. Bars indicate relative mean ± s.d., $n = 3$ biological replicates. ANOVA modelling relative area under the curve with IFN and siRNA as explanatory variables: IFN, $F_{(1,12)} = 23,829.4$, $p < 0.0001$; siRNA, $F_{(2,12)} = 211.3$, $p < 0.0001$; antagonising interaction, $F_{(2,12)} = 211.3$, $p < 0.0001$. All data are representative of two independent experiments. a.u., arbitrary units. Multiple comparisons: ****$p < 0.0001$

cells, LANA accumulation was detected in the majority of cells after overnight infection with KSHV (Supplementary Fig. 3a). A549-MxB cells exhibited only 10–17% of the average signal intensity measured in the respective control condition (Fig. 3b and Supplementary Fig. 3a). Similarly to our results in the flow cytometry-based assay (Fig. 3a), endogenous LANA expression was rescued upon *MX2* silencing in A549-MxB cells (50–92% of the respective control), emphasising the specificity of MxB in restricting KSHV (Fig. 3b and Supplementary Fig. 3a). GFP and LANA signals measured in parallel revealed a strong correlation (Spearman's $\rho = 0.787$, $p < 0.0001$, Supplementary Fig. 3a), which

is consistent with the notion that GFP under transcriptional control of the eukaryotic translation elongation factor 1α (*EEF1A1*) promoter can be used as a marker of KSHV infection[51]. KSHV lytic gene expression, as measured by RFP expression under the control of the KSHV lytic polyadenylated nuclear RNA (*PAN*) promoter, was hardly detectable in A549 cells at 48 h p.i., suggesting a predominantly latent infection of these epithelial cells. Since KSHV also exhibits a natural tropism for endothelial cells[54], we tested MxB-dependent restriction of KSHV in primary human umbilical vein endothelial cells (HUVECs). We initially noted that HUVECs induced type I IFN massively and invariantly

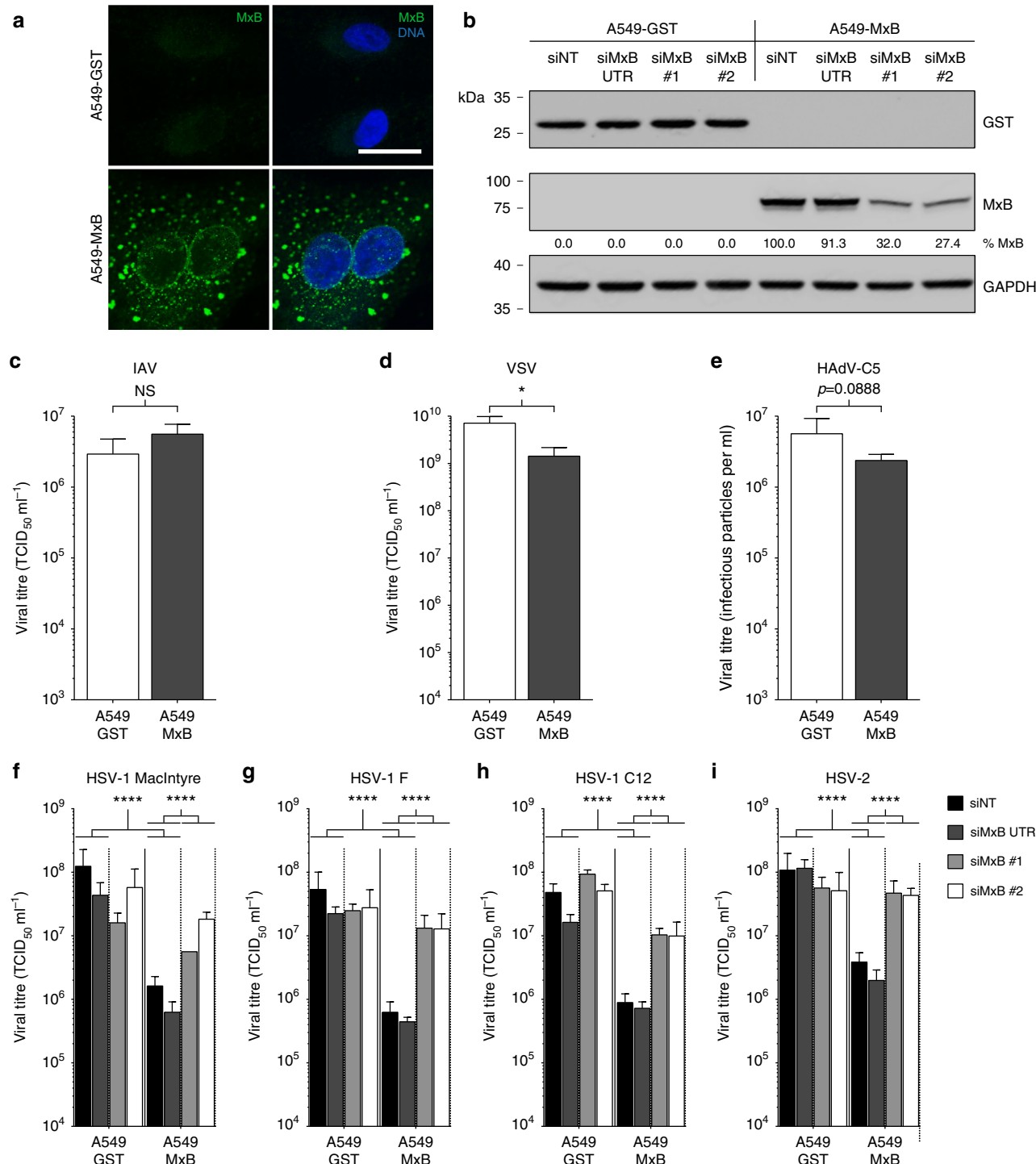

in response to transfection of nucleic acids or transduction with lentiviral particles. To prevent this effect, we used co-transfection of siRNA against IFN regulatory factor 9 (*IRF9*) together with non-targeting siRNA or siRNA against *MX2* and subsequently transduced the cells with lentiviral vectors encoding either GST or MxB (Supplementary Fig. 3b). Infection of HUVECs with KSHV in this setting revealed a modest MxB-dependent restriction (26% less infected cells as compared to the control) that was fully released upon *MX2* silencing (Supplementary Fig. 3c). KSHV lytic reporter gene expression remained only marginally above background and was therefore omitted from the analysis. Thus, MxB inhibits KSHV early in infection, most likely at the stage of virus entry.

**MxB inhibits accumulation of HSV-1 gene products**. MxB expression blocks HIV-1 replication at an early step before integration of proviral DNA into the host genome[13]. We therefore asked whether MxB exerts a gatekeeper function by restricting nuclear entry of herpesviral DNA. To this end, we first tested the expression of HSV-1 immediate-early and early genes at 4.5 h p.i. We carried out quantitative reverse transcription PCR (RT-qPCR) analyses with RNA isolated from A549-GST and A549-MxB cells infected with HSV-1 strain MacIntyre, F or C12. In the presence of MxB, mRNA accumulation of immediate-early genes *RL2* and *RS1* (encoding ICP0 and ICP4, respectively) and early gene *UL29* (encoding ICP8) was reduced 2–3-fold in the strains MacIntyre and F (Fig. 4a, b) and 5–7.5-fold in the recombinant strain C12 (Fig. 4c). The apparent differences between strain C12 and the two natural strains may be explained by different kinetics in life cycle progression, enhanced sensitivity to MxB or a combination of both.

Our results from the RT-qPCR analyses were reflected in significantly and consistently reduced de novo expression of immediate-early proteins ICP0 and ICP4 and early protein ICP8 in A549-MxB cells infected with HSV-1 strain MacIntyre or F (Fig. 4d, e). As a consequence, we also observed a strong inhibition of VP5 synthesis in both strains (Fig. 4d, e).

**MxB blocks HSV-1 genome uncoating and nuclear translocation**. After internalisation of HSV-1 particles into the cytoplasm, tegument proteins begin to dissociate from the capsid, allowing the major capsid protein VP5 to become accessible to specific antibodies[55]. To study the effect of MxB on the exposure of epitopes on the viral capsid during entry, we infected A549-GST or A549-MxB cells with HSV-1 strain F at high MOI in the presence of the protein synthesis inhibitor cycloheximide. Using

immunofluorescence assays with an antibody directed against VP5, we monitored the appearance of exposed VP5 epitopes in the cytoplasm during the first 120 min of infection. The HSV-1 VP5 signal remained undetectable during the first 60 min of infection, and then gradually increased at later time points. The number of VP5-specific foci in the cytoplasm likely represented individual capsids or capsid assemblies, and was not altered in cells expressing MxB, suggesting that MxB did not inhibit viral tegument dissociation (Fig. 5a). We then asked whether MxB would interfere with trafficking of HSV-1 capsids to the nuclear envelope and/or translocation of genomic DNA into the nucleus. We used a combined immunofluorescence and click chemistry approach to simultaneously detect individual HSV-1 capsids in the perinuclear region and uncoated, condensed HSV-1 genome foci in the nucleus[56]. EdC genome-labelled HSV-1 was added to A549-GST and A549-MxB cells in the presence of cycloheximide to prevent early gene transcription and replication[57]. Cycloheximide has previously been shown not to interfere with HSV-1 genome uncoating and translocation to the nucleus[56]. We assessed the accumulation of capsids in the perinuclear region with an antibody directed against purified DNA-containing HSV-1 capsids (anti-HC). Intriguingly, in the presence of MxB, we observed 51% less accumulation of capsids at the nucleus 30 min p.i., and 67% less accumulation at 120 min p.i. compared to the control (Fig. 5b). In line with this, nuclear import of uncoated HSV-1 genomes was strongly reduced in A549-MxB cells as compared to A549-GST cells, where it was detectable at 30 min p.i. and peaked at 120 min p.i. (Fig. 5c). These observations were corroborated by qPCR measurements of the viral genome in the nuclear fraction of A549-MxB cells in presence of cycloheximide. For the HSV-1 strains MacIntyre, F and C12, cells with a silenced *MX2* gene exhibited a 1.4–2.2-fold increase of viral genomic DNA in the nuclear fraction compared to the control conditions (Supplementary Fig. 4a). The efficiency of *MX2* silencing and the efficiency of subcellular fractionation were validated in parallel samples using qPCR of the host gene glyceraldehyde-3-phosphate dehydrogenase (*GAPDH*) in the nucleus, and immunoblots of α-tubulin and histone H3 as cytoplasmic and nuclear markers, respectively (Supplementary Fig. 4b). In control siRNA-treated cells, MxB protein was mostly associated with nuclear fractions, consistent with previously published results[14].

If MxB exerted a gatekeeper function at the NPCs or in the distant cytoplasm, thus interfering with uncoating and/or nuclear translocation of HSV-1 genomic DNA, we would expect more DNA-containing capsids in the cytoplasm of MxB-expressing cells compared to control cells. To test this hypothesis, we

**Fig. 2** Ectopic expression of MxB inhibits lytic replication of HSV-1 and HSV-2. **a** A549 human lung adenocarcinoma cells were engineered to stably express glutathione *S*-transferase (GST) or human MxB. A549-GST and A549-MxB cell lines were generated from clones with high gene expression. MxB protein expression and intracellular localisation was assessed by immunostaining. Nuclei were stained with Hoechst 33342. Scale bar, 20 μm. **b** A549-GST and A549-MxB cells were transfected with non-targeting siRNA (siNT), siRNA targeting the 3′-UTR of endogenous *MX2* (siMxB UTR) or two different siRNAs targeting the coding sequence of endogenous and overexpressed *MX2* (siMxB #1, siMxB #2). At 48 h post transfection, cells were lysed for immunoblot analysis of GST and MxB protein expression and *MX2* silencing efficiency. GAPDH served as a loading control. **c–e** A549-GST and A549-MxB cells were infected with IAV strain WSN (MOI = 0.5) for 24 h (**c**), VSV serotype Indiana (MOI = 0.5) for 16 h (**d**) or HAdV-C5 strain wt300 (MOI = 5) for 48 h (**e**). Viral output was measured by $TCID_{50}$ assay (**c**, **d**) or immunofluorescence assay (**e**). Bars indicate mean ± s.d., $n = 3$–4 biological replicates. Unpaired two-tailed Student's *t* tests with log-transformed titres: NS $p ≥ 0.05$; *$p < 0.05$. **f–i** A549-GST and A549-MxB cells were transfected with the indicated siRNAs. At 48 h post transfection, cells were infected with HSV-1 strain MacIntyre or F (MOI = 0.05) for 24 h (**f**, **g**), recombinant strain C12 (MOI = 0.5) for 24 h (**h**) or HSV-2 strain G (MOI = 0.05) for 48 h (**i**). Viral output was measured by $TCID_{50}$ assay. Data are representative of three independent experiments. Bars indicate mean ± s.d., $n = 3$ biological replicates. ANOVAs modelling log-transformed titre with cell line and siRNA as explanatory variables: HSV-1 MacIntyre: cell line, $F_{(1,16)} = 86.442$, $p < 0.0001$; siRNA, $F_{(3,16)} = 6.863$, $p = 0.003489$; antagonising interaction, $F_{(3,16)} = 12.34$, $p = 0.000196$. HSV-1 F: cell line, $F_{(1,16)} = 49.065$, $p < 0.0001$; siRNA, $F_{(3,16)} = 6.653$, $p = 0.00399$; antagonising interaction, $F_{(3,16)} = 8.102$, $p = 0.00166$. HSV-1 C12: cell line, $F_{(1,16)} = 333.01$, $p < 0.0001$; siRNA, $F_{(3,16)} = 46.05$, $p < 0.0001$; antagonising interaction, $F_{(3,16)} = 10.47$, $p = 0.000471$. HSV-2: cell line, $F_{(1,16)} = 54.133$, $p < 0.0001$; siRNA, $F_{(3,16)} = 5.455$, $p = 0.00891$; antagonising interaction, $F_{(3,16)} = 18.121$, $p < 0.0001$. Multiple comparisons: ****$p < 0.0001$

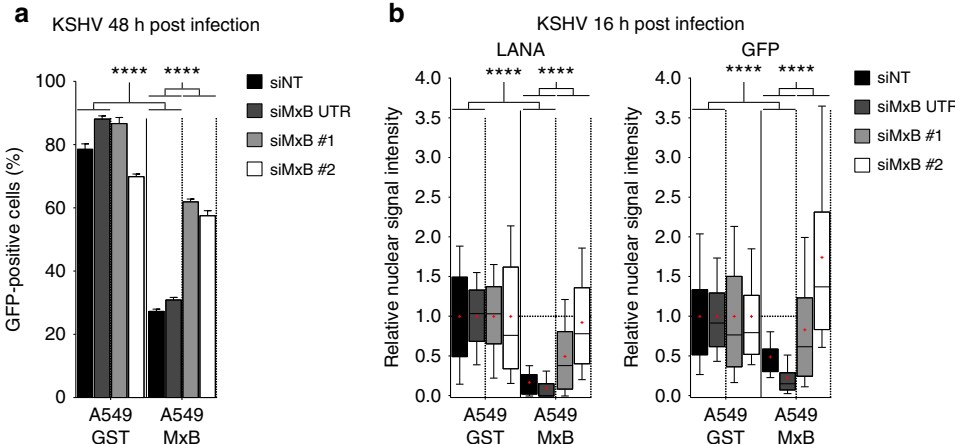

**Fig. 3** Ectopic expression of MxB inhibits KSHV latent infection and protein expression. **a** A549-GST and A549-MxB cells were transfected with the indicated siRNAs. At 48 h post transfection, cells were infected with recombinant KSHV expressing GFP (MOI = 0.5) for 48 h. Viral output was measured by flow cytometry. Data are representative of three independent experiments. Bars indicate mean ± s.d., n = 3 biological replicates. ANOVA modelling arcsine-transformed fraction of GFP-positive cells with cell line and siRNA as explanatory variables: cell line, $F_{(1,16)} = 3920.2$, $p < 0.0001$; siRNA, $F_{(3,16)} = 227.1$, $p < 0.0001$; antagonising interaction, $F_{(3,16)} = 325.2$, $p < 0.0001$. **b** A549-GST and A549-MxB cells were transfected as in **a**, but then infected with recombinant KSHV expressing GFP (MOI = 0.5) for 16 h. Intranuclear levels of endogenous latency-associated nuclear antigen (LANA, left panel) and GFP reporter (right panel) were measured using immunofluorescence assay. Data are representative of two independent experiments, showing nuclear signal intensity in A549-MxB cells relative to A549-GST cells for each group. Boxes depict median, 25th and 75th percentile with mean marked with a '+'. Whiskers denote the 10th and 90th percentile. For each comparison, 354 cells were randomly selected. ANOVAs modelling log-transformed relative signal intensity with cell line and siRNA as explanatory variables: LANA: cell line, $F_{(1,2824)} = 1283.6$, $p < 0.0001$; siRNA, $F_{(3,2824)} = 152.4$, $p < 0.0001$; antagonising interaction, $F_{(3,2824)} = 218.5$, $p < 0.0001$. GFP: cell line, $F_{(1,2824)} = 300.4$, $p < 0.0001$; siRNA, $F_{(3,2824)} = 225.5$, $p < 0.0001$; antagonising interaction, $F_{(3,2824)} = 256$, $p < 0.0001$. Multiple comparisons: ****$p < 0.0001$

performed transmission electron microscopy (TEM) to image the incoming HSV-1 particles containing viral DNA (full capsids with electron-dense material) or lacking viral DNA (empty particles). We infected MxB-expressing and *MX2*-silenced cells with HSV-1 strain MacIntyre for 3 h in the presence of cycloheximide and processed the samples for TEM analysis (Fig. 6). In parallel, *MX2* silencing efficiency was determined by immunoblot (Supplementary Fig. 7c). The number of HSV-1 capsids in the cytoplasm or in multivesicular bodies (MVBs) was determined in ultra-thin sections across 16 control siRNA-treated and 22 *MX2*-specific siRNA-treated cells (Fig. 6e). The cytoplasm of MxB-expressing cells exhibited mostly full capsids (Fig. 6a), while *MX2*-silenced cells contained primarily empty capsids (Fig. 6b). The plasma membrane and MVBs of both MxB-expressing and *MX2*-silenced cells displayed virions containing full capsids in approximately similar amounts (Fig. 6c, d), representing quasi-equivalent inocula. Collectively, these data suggest that MxB interferes with HSV-1 entry at a step after tegument dissociation but before viral genome uncoating and translocation into the nucleus.

**HSV-1 restriction by MxB requires an intact GTPase domain.** Whereas the antiviral function of MxA depends on the GTP-binding and GTP-hydrolysing activities, the GTPase activity of MxB is apparently not required to inhibit HIV-1 replication[9,10,13,19,20]. Determinants of anti-lentivirus activity were instead found within the amino-terminal region of MxB[17,19,58,59].

To elucidate whether the amino terminus of MxB and/or its GTPase function were necessary for HSV-1 restriction, we made use of the herpesvirus-inducible luciferase reporter plasmid pGL-T9G[60]. The construct was transiently transfected into HeLa cells together with increasing amounts of expression plasmids encoding GST, wild-type MxB, the short isoform of MxB (MxB Δ1–25), an MxB mutant where a specific residue that has been

under strong positive selection in Old World monkeys and hominoids was converted to a residue found in New World monkeys (MxB G51Q)[21], or an MxB mutant with a point mutation in the globular (G) domain that abrogates the GTP-hydrolysing activity (MxB T151A)[47,61]. We reasoned that a transient expression system would allow us to study the behaviour of different MxB variants in a dose-dependent manner without possible adverse effects of clonal selection in stable overexpressing systems. Infection of GST-expressing cells with HSV-1 strain MacIntyre led to a 100-fold induction of the herpesvirus-specific luciferase reporter, irrespectively of the amount of expressed transgene (Fig. 7a). The reporter signal augmented with time and increasing MOI and could be antagonised by adding the HSV-specific DNA polymerase inhibitor phosphonoacetic acid, demonstrating that the reporter activity was entirely dependent on progressive HSV-1 infection (Supplementary Fig. 6c–e). Furthermore, a reporter that expresses luciferase under the control of a cellular promoter was not affected by increasing doses of GST, wild-type MxB or mutants thereof (Supplementary Fig. 6b). For wild-type MxB-overexpressing cells or MxB G51Q-overexpressing cells infected with HSV-1 strain MacIntyre, we observed a significant dose-dependent inhibition of the herpesvirus reporter as compared to the GST control. However, this effect was not seen in cells expressing either the short isoform or the GTPase-deficient form of MxB (Fig. 7a). Similar experiments with HSV-1 strain F and HSV-2 yielded comparable results (Supplementary Fig. 6a).

We then tested whether there was a link between anti-herpesvirus activity and subcellular localisation of MxB, as previously suggested for MxB in HIV-1 restriction[19]. The characteristic perinuclear localisation of MxB was observed in both MxB mutants but not in the short isoform lacking the first 25 amino acids (Fig. 7b).

Finally, to control for antiviral activities of the different MxB variants, we transfected HeLa cells with a similar panel of

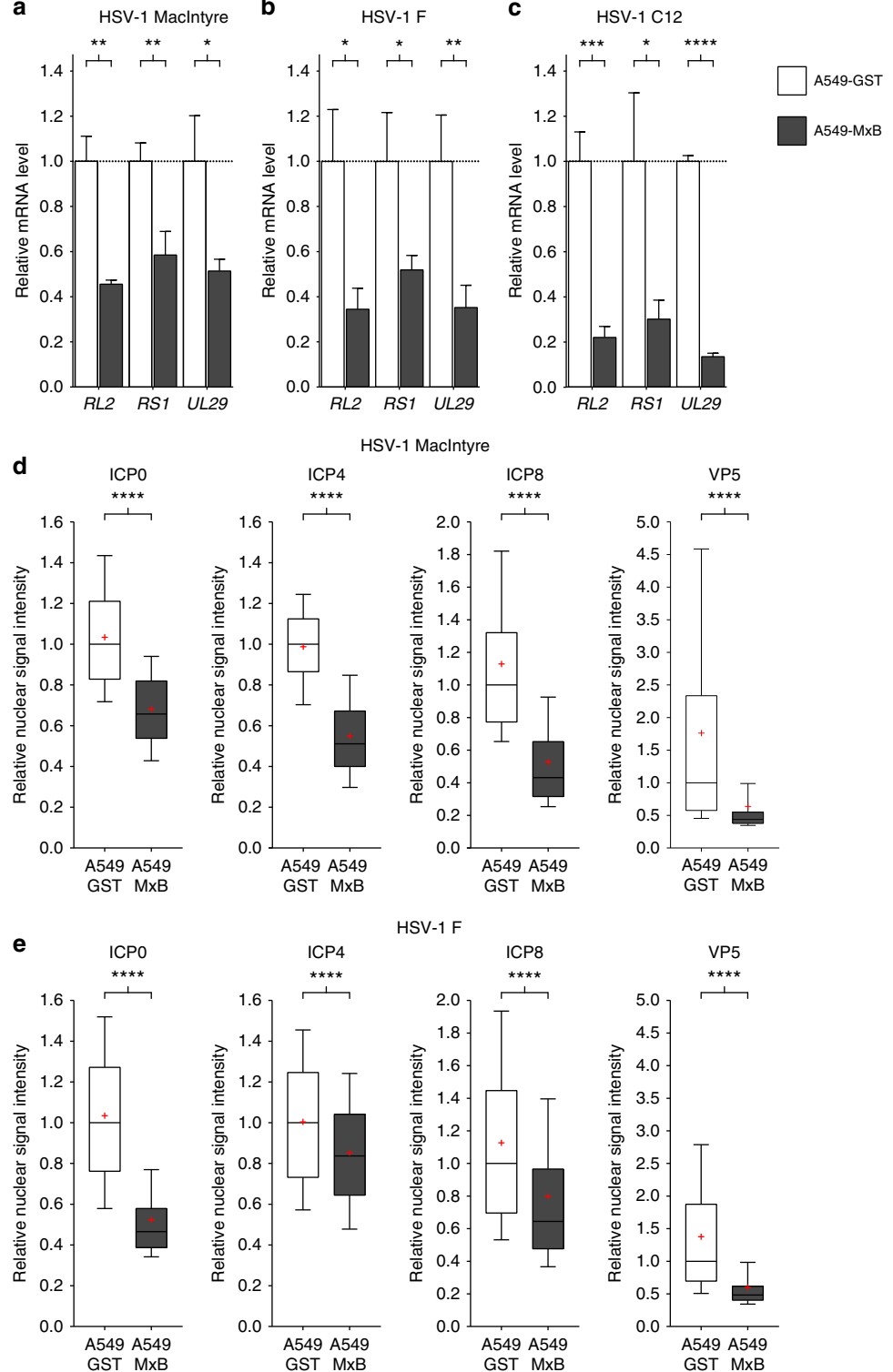

**Fig. 4** MxB inhibits accumulation of HSV-1 immediate-early transcripts and proteins. **a–c** A549-GST and A549-MxB cells were infected with HSV-1 strain MacIntyre (**a**), F (**b**) or C12 (**c**) (MOI = 0.5) for 4.5 h. Cells were lysed and total RNA extracted for cDNA synthesis and RT-qPCR analysis of the indicated transcripts. Bars indicate relative mean ± s.d., $n = 3$ biological replicates averaged from three technical replicates for each sample. Unpaired two-tailed Student's $t$ tests: *$p < 0.05$, **$p < 0.01$, ***$p < 0.001$ and ****$p < 0.0001$. **d**, **e** A549-GST and A549-MxB cells were infected with HSV-1 strain MacIntyre (**d**) or F (**e**) (MOI = 20) and fixed at 3 h (ICP0, ICP4) or 4 h (ICP8, VP5) post infection. HSV-1 de novo protein expression was assessed by immunostaining with antibodies directed against the indicated proteins, followed by confocal microscopy. Data show nuclear signal intensity in A549-MxB cells relative to A549-GST cells for each group. Boxes depict median, 25th and 75th percentile with mean marked with a '+'. Whiskers denote the 10th and 90th percentile. For each comparison, 336 cells were randomly selected. Unpaired two-tailed Wilcoxon's rank sum test: ****$p < 0.0001$. All data are representative of three independent experiments

expression vectors but infected with the VSV-pseudotyped, HIV-1-based luciferase reporter virus NL-Luc. In accordance with previous findings[10], amino-terminal residues that are found in the long but not in the short isoform of MxB were indispensable for the interference with HIV-1 infection, but the GTPase function could be abrogated without losing the anti-HIV-1 effect (Supplementary Fig. 5a).

## Discussion

Type I IFN interferes with herpesvirus replication in both epithelial and neuronal cells[62]. Several IFN-induced effector proteins, including PKR and OAS, have been implicated in restricting

herpesvirus infection so far[63,64], but the extent of their respective contributions is difficult to assess due to the elaborate evasion strategies that the herpesviruses evolved[41,42].

Here we demonstrate that MxB plays a critical role in inhibiting several members of the *Alphaherpesvirinae* and *Gammaherpesvirinae* subfamilies. In agreement with our results, murine gammaherpesvirus 68 has already been implicated as a target of MxB in a screen for IFN-induced antiviral factors[50]. The contribution of MxB to the type I IFN-induced restriction of HSV-1 is highly significant. Out of the 3-log$_{10}$ inhibition imposed by IFN treatment, 0.9–1.4 log$_{10}$ could be attributed to the contribution of *MX2* (Fig. 1b, c). These observations suggest that herpesviruses are yet to evolve effective strategies to evade the activity of MxB.

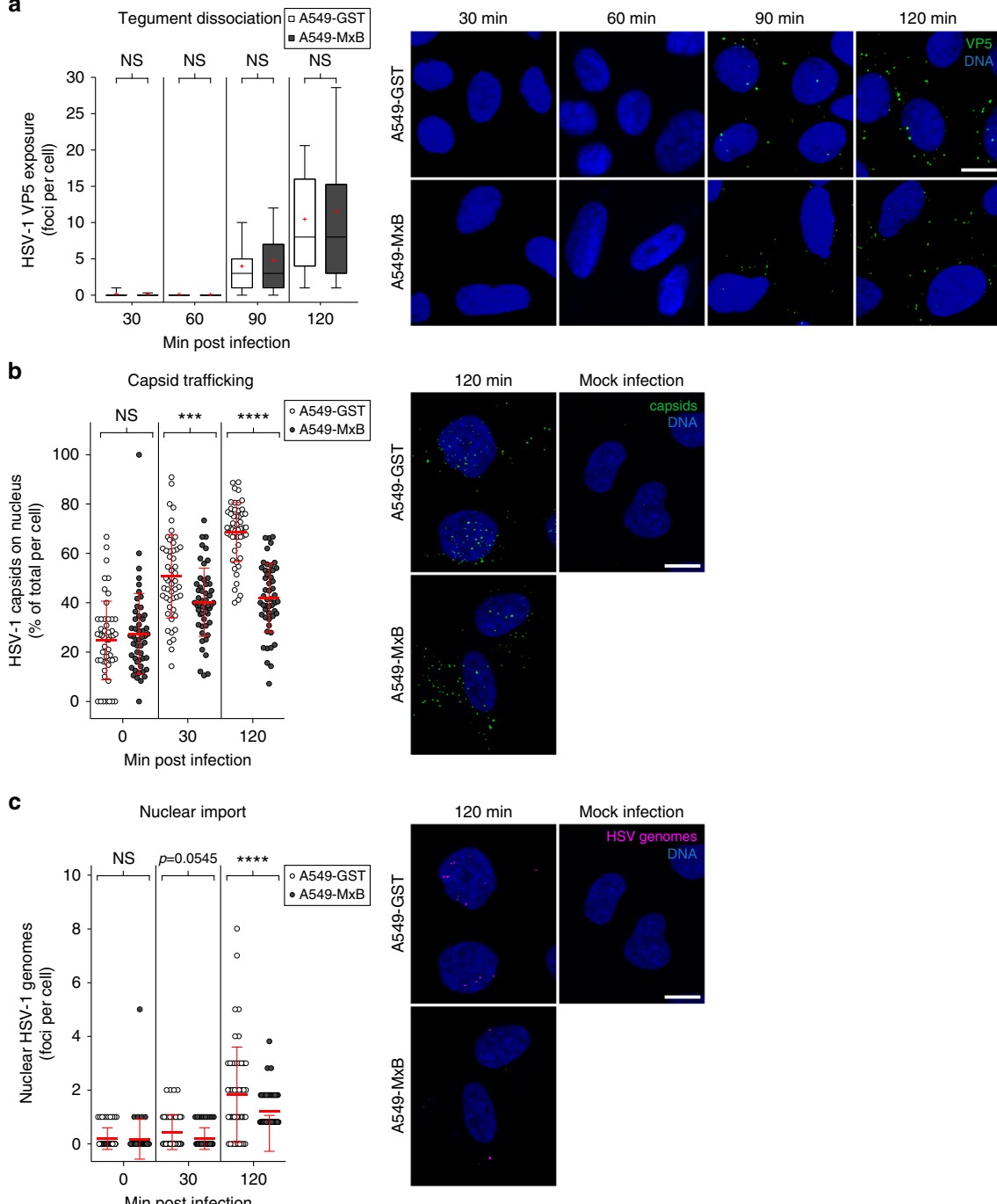

Remarkably, the contribution of MxB to the restriction of HSV-1 is at least as pronounced as the one of PKR[63], since silencing of *PKR* and *MX2* resulted in comparable rescue of reporter expression of the C12 strain (compare Fig. 1d with Supplementary Fig. 1c). Silencing of ectopically expressed MxB with siRNA against the coding region of *MX2* led to a complete or near-complete relieve of herpesvirus restriction (Figs. 2f–i and 3a, b). These findings indicate that MxB directly interferes with herpesvirus replication and not indirectly, for example, by inducing antiviral effector proteins. However, we cannot exclude other cell-intrinsic or IFN-related factors contributing to the activity of MxB against herpesviruses, since Vero cells (monkey-derived cells deficient in the synthesis of type I IFN) exhibited a lower degree of inhibition compared to A549 cells (compare Supplementary Fig. 2a with Supplementary Fig. 2e), despite similar expression levels of human MxB (Fig. 2b and Supplementary Fig. 2d). Alternatively, the differences observed in the two cell types may be explained by differences in the cell-cycle stage at the time of infection, given the role of MxB in cell-cycle progression and the negative effect of S-phase inhibition on the antiviral potency against HIV-1[10,47].

Available evidence suggests that MxB is rather selective with respect to its antiviral targets (see Ref. [13] and our study). Primate lentiviruses and herpesviruses are unrelated to each other, yet have similar nuclear entry mechanisms at least with respect to using the NPC as a gateway into the nucleus[38]. They are both targets of MxB, whereas MxB does not affect other RNA viruses that import their genome into the nucleus via the NPC, such as IAV (see Ref. [12] and Fig. 2c). This supports the view that MxB does not act by controlling transport processes through the NPC per se[47]. Indeed, viruses that replicate independently of nuclear functions, such as the rhabdovirus VSV (see Ref. [50] and Fig. 2d) and hepatitis C virus (HCV)[65], may also be targeted by MxB in the cytoplasm. We observed a minor reduction of HAdV-C5 multi-cycle infection in A549-MxB cells (Fig. 2c) and a pronounced inhibition of single-round infection (Supplementary Fig. 5c). However, this infection could not be rescued by *MX2*-specific siRNAs (Supplementary Fig. 5d), suggesting MxB-independent effects or indirect effects of MxB on HAdV-C5 infection. For instance, the continued expression of MxB may induce IFN and/or production of other ISGs restricting HAdV-C5. MxB-induced gene products may be long lasting and resist *MX2* silencing, yet act as auxiliary factors against HAdV-C5. Such a scenario has been discussed for the MxB paralog MxA in HCV restriction[66]. The lack of such auxiliary factors against HIV-1 may explain the observation that endogenous MxB does not contribute to the type I IFN-mediated inhibition of HIV-1 in

certain human cell lines[67]. All in all, it is debatable whether MxB directly restricts HIV-1 infection. MxB has been reported to bind to tubular recombinant HIV-1 capsid structures in vitro[15]. However, mutations in the HIV-1 capsid rendering HIV-1 resistant to MxB do not affect the interaction of the capsid with MxB, suggesting indirect effects in restricting HIV-1[15]. It remains to be elucidated whether MxB targets herpesvirus capsids and/or components of the NPC such as Nup358 or Nup214, which are implicated in nuclear import of both HIV-1 proviral DNA and HSV-1 genomic DNA[68–71].

MxB inhibits herpesviruses at an early step in the life cycle. Our data indicate that MxB acts at a step after virus entry and tegument dissociation from the capsid (Fig. 5a) but prior to uncoating of viral DNA from the capsid (Fig. 6) and nuclear import of viral DNA (Fig. 5c, Supplementary Fig. 4a). As a consequence, consecutive steps such as accumulation of immediate-early HSV-1 transcripts and proteins are reduced (Fig. 4).

Collectively, these observations are in agreement with the hypothesis of a cytoplasmic gatekeeper function of MxB against herpesviruses. It is possible that the block occurs directly at the NPC, as suggested by the subcellular localisation of MxB (Figs. 1a, 2a and 7b). We cannot rule out, however, that MxB blocks the microtubule-directed transport of HSV-1 capsids to the nuclear pores, or another licencing step in the cytoplasm that renders the capsids competent for DNA uncoating, or capsid docking to the NPC. The latter scenario may be supported by the observation that MxB-expressing cells had low numbers of capsids near the nucleus (Fig. 5b).

In addition to blocking herpesvirus entry, it is possible that MxB has antiviral activity also at later time points in the viral life cycle, for example, during exit of newly assembled capsids from the nucleus. While HSV protein expression could be fully rescued when *MX2* was silenced in A549-MxB cells, a complete rescue of virus titres was not always achieved (compare Fig. 2f–i with Supplementary Figs. 2a and 7b). Thus, we cannot exclude that MxB impacts the production of mature, infectious virions at multiple levels.

Examination of structural and functional features of MxB required for inhibiting herpesvirus replication revealed that residues within the amino-terminal 25 amino acids of the long isoform and the GTPase function of MxB were critical for MxB activity against HSV-1 and HSV-2 (Fig. 7a and Supplementary Fig. 6a). This is reminiscent of the MxB paralog MxA, which depends on a functional GTPase to inhibit many RNA and DNA viruses in the cytoplasm[13]. Intriguingly, the perinuclear association of the MxB T151A mutant was stronger than wild-type MxB (Fig. 7b), consistent with the hypothesis that the site of MxB

**Fig. 5** MxB inhibits perinuclear association of HSV-1 capsids and nuclear translocation of viral genomes. **a** A549-GST and A549-MxB cells were infected with HSV-1 strain F in the presence of cycloheximide. Cells were fixed at the indicated time points post infection. HSV-1 capsids were stained with anti-VP5 antibody and nuclei were stained with Hoechst 33342. Samples were analysed by confocal microscopy. Data represent the number of VP5-exposed dots per cell. Boxes depict median, 25th and 75th percentile with mean marked with a '+'. Whiskers denote the 10th and 90th percentile. For each comparison, 166 cells were randomly selected. Kruskal–Wallis test with cell line and time as explanatory variables: cell line, $\chi^2 = 0.0033671$, df = 1, $p = 0.9537$; time, $\chi^2 = 896.1$, df = 3, $p < 0.0001$. Pairwise Wilcoxon's rank sum tests: NS $p = 1.00$. Data are representative of two independent experiments. Right panels show representative images. Scale bar, 10 μm. **b, c** A549-GST and A549-MxB cells were mock-infected or infected with EdC-labelled HSV-1 strain F (purified by density gradient centrifugation, approx. 40 particles per cell) in the presence of cycloheximide and fixed at the indicated time points post infection. Nuclei were stained with DAPI, capsids were stained with anti-HC antibody and free genome foci were visualised using click chemistry. Cells were analysed by confocal microscopy with z-series image acquisition and projection along the z-axis. Individual measurements with mean ± s.d. are shown. Data are representative of three independent experiments. For each comparison, 51 cells were randomly selected. Representative images show maximum projections. Scale bar, 10 μm. **b** Percentage of capsids per cell coinciding with the nuclear mask. ANOVA modelling fraction of capsids on nucleus with cell line and time as explanatory variables: cell line, $F(1,300) = 45.3$, $p < 0.0001$; time, $F(2,300) = 101.2$, $p < 0.0001$; antagonising interaction, $F(2,300) = 24.91$, $p < 0.0001$. Multiple comparisons: NS $p \geq 0.05$, ***$p < 0.001$ and ****$p < 0.0001$. **c** Number of HSV genomes per cell coinciding with the nuclear mask. Kruskal–Wallis test with cell line and time as explanatory variables: cell line, $\chi^2 = 23.054$, df = 1, $p < 0.0001$; time, $\chi^2 = 45.162$, df = 2, $p < 0.0001$, antagonising interaction, $\chi^2 = 81.936$, df = 5, $p < 0.0001$. Pairwise Wilcoxon's rank sum tests: NS $p \geq 0.05$; ****$p < 0.0001$

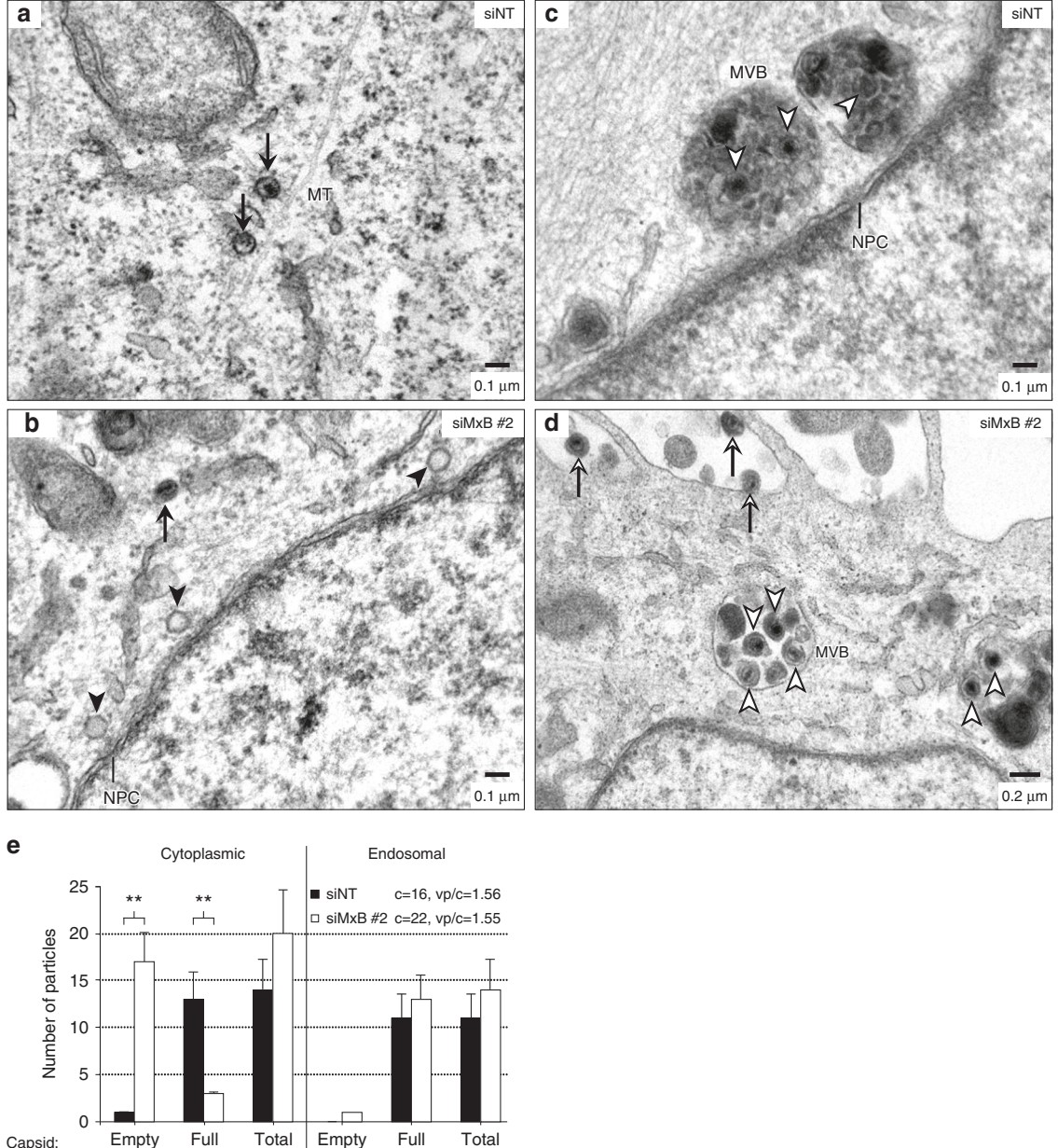

**Fig. 6** MxB inhibits HSV-1 genome uncoating. **a–d** A549-MxB cells were transfected with the indicated siRNAs. At 60 h post transfection, cells were infected with HSV-1 strain MacIntyre (MOI = 500) in the presence of cycloheximide. Cells were fixed 3 h post infection and subjected to TEM analysis. Black arrows: full capsids; black arrowheads: empty capsids; open arrows: extracellular virions at the plasma membrane; open arrowheads: virions in multivesicular bodies (MVB). MT, microtubule; NPC, nuclear pore complex. **e** Subcellular quantification of empty and full capsids in the cytoplasm and endosomal compartments depicting the mean number of virus particles (vp) in analysed cells (c) + s.e.m. as derived from unpaired two-tailed Student's $t$ tests. $**p < 0.01$. vp/c, average number of virus particles analysed per cell

enzymatic activity is close to this location[47]. It is possible that MxB associates with the nuclear envelope and thereby interferes with herpesvirus uncoating. However, any perinuclear localisation of MxB would not be sufficient, since the non-restricting GTPase-deficient mutant MxB T151A was also found in the perinuclear region.

The amino acid residue at position 51 in simian MxB was shown to be under strong positive selection in Old World monkeys and hominoids, with a glycine emerging as a fixed mutation in this primate lineage[21]. We thus tested whether G51 of human MxB was involved in herpesvirus restriction by mutating the glycine to glutamine as seen in New World

monkeys. However, this conversion did not alter the activity of MxB against HSV-1 (Fig. 7a), which leaves the origin of selection pressure at this residue unknown. We suggest that in catarrhine species, MxB restricts pathogens beyond herpesviruses and lentiviruses[21].

Evidently, the GTP hydrolysis-mediated conformational change of MxB[47] is indispensable for its activity against herpesviruses, while the amino-terminal region upstream of the G domain of MxB appears to be sufficient for the inhibition of HIV-1[19]. This points at distinct inhibitory mechanisms of MxB against HSV-1 and HIV-1. Nevertheless, the fact that the amino terminus of MxB is important for restricting both HSV-1 and HIV-1, and

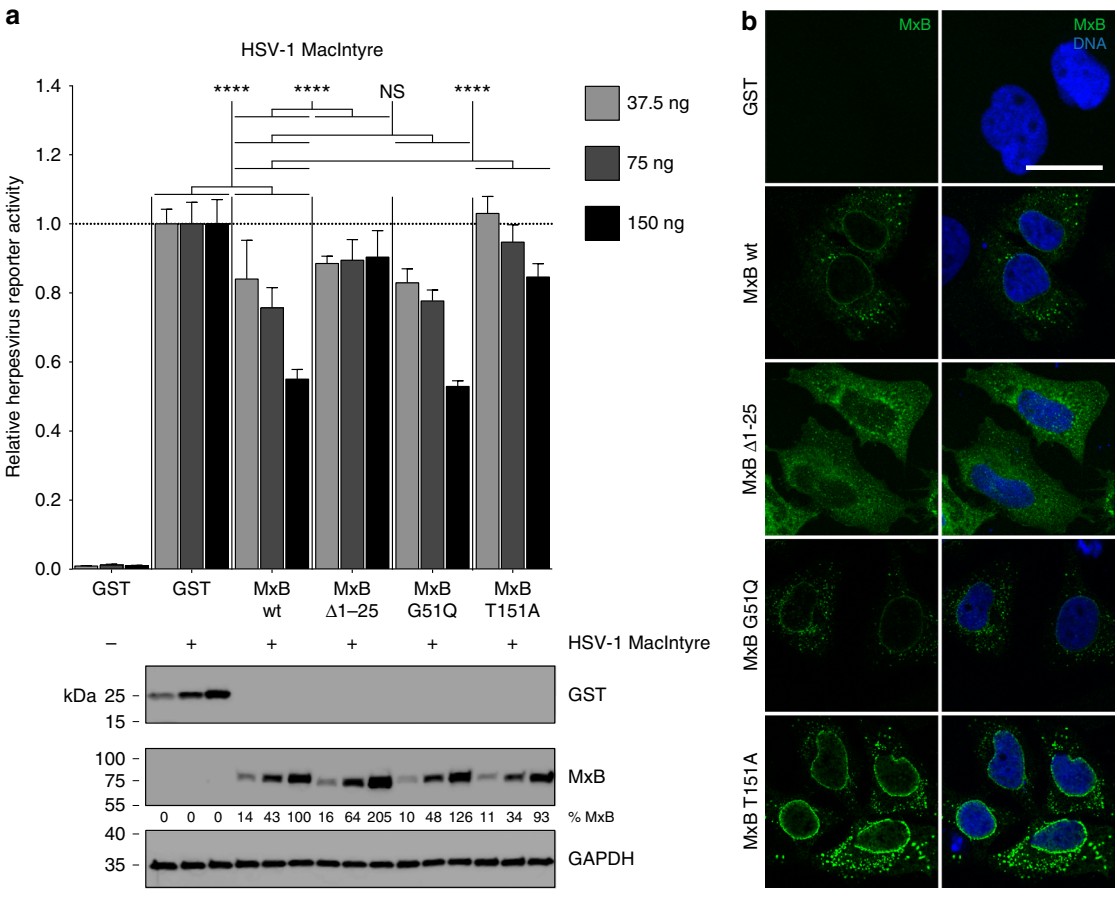

**Fig. 7** HSV-1 restriction by MxB is specified by its amino terminus and requires GTPase function. **a** HeLa cells were transiently transfected with pGL-T9G reporter plasmid together with increasing amounts of expression plasmids encoding the indicated proteins. At 24 h post transfection, cells were mock-infected or infected with HSV-1 strain MacIntyre (MOI = 0.5). Virus infection efficiency was determined by luciferase assay at 32 h post infection and is presented relative to the respective conditions with GST expression. Luciferase activity from all conditions is normalised to mock-infected cells. Data are representative of three independent experiments. Bars indicate mean ± s.d., $n = 3$ biological replicates. ANOVA modelling relative reporter activity with protein variant and dose as explanatory variables: protein variant, $F(4,30) = 51.311$, $p < 0.0001$; dose, $F(2,30) = 29.844$, $p < 0.0001$; synergistic interaction, $F(8,30) = 6.407$, $p < 0.0001$. Multiple comparisons: NS $p \geq 0.05$; ****$p < 0.0001$. GST and MxB protein expression was controlled by immunoblot analysis using pooled samples for each condition. GAPDH served as loading control. **b** HeLa cells were transiently transfected with expression plasmids encoding the indicated proteins. At 24 h post transfection, cells were fixed and MxB protein expression and intracellular localisation was assessed by immunostaining. Nuclei were stained with Hoechst 33342. Scale bar, 20 μm

that both viruses are blocked at an early post-entry step, at or before uncoating of viral genomes, argue for a common direct or indirect cytoplasmic gatekeeper function of MxB.

## Methods

**Cell lines**. Human glioblastoma cells T98G (CRL-1690), human lung adeno-carcinoma cells A549 (CRM-CCL-185), African Green Monkey kidney epithelial cells Vero E6 (CRL-1586), HEK-293T (CRL-11268) and HeLa cells (CCL-2) were purchased from ATCC. Cells were cultured in growth medium (Dulbecco's modified Eagle's medium (DMEM, high glucose) supplemented with 10% foetal bovine serum (FBS), 200 μM GlutaMAX, 100 IU/ml penicillin and 100 μg/ml streptomycin (Life Technologies)) at 37 °C and 5% $CO_2$. Primary HUVECs (CRL-1730, ATCC, kindly provided by Mirco Ponzoni, IRCCS Istituto G. Gaslini, Genova, Italy) were cultured on plates pre-coated with collagen derived from rat tail tendon (Merck) in HUVEC medium (EBM-2 endothelial growth basal medium (CC-3156, Lonza) supplemented with EGM-2 Endothelial Growth SingleQuot Kit Supplement and Growth Factors (CC-4176, Lonza)) at 37 °C and 5% $CO_2$. For the generation of stably transduced A549 and Vero cell lines, full-length *Schistosoma japonicum* GST (*GSTM1*, accession no. FN315690.1) and full-length *Homo sapiens MX2* (accession no. nM_002463.1) complementary DNAs (cDNAs) were cloned into the lentiviral expression vector pLVX-IRES-Puro (Clontech) using the *Xho*I and *Xba*I restriction sites to obtain pLVX-GSTM1-IRES-Puro or the *Xho*I and *Not*I restriction sites to obtain pLVX-MX2-IRES-Puro. Lentiviral particles were produced in HEK-293T cells with pCMVR8.91-Gag-Pol and pVSV-G (Clontech)

together with the lentiviral expression constructs. A549 and Vero cells were transduced with the packaged expression vectors and clones with high gene expression were selected and maintained with 5 μg/ml puromycin (Invivogen). As a quality control, the identity of the stable A549 and Vero clones was confirmed by sequencing.

**Plasmids**. For transient overexpression experiments, full-length *GSTM1* and *MX2* cDNAs plus the cDNA fragment encoding the short isoform of MxB (MxB Δ1–25, base pairs 76–2148 of *MX2* cDNA) were cloned into pCDNA3.1-Neo(+) expression vector (Invitrogen) using the *Not*I and *Xba*I restriction sites. The *MX2(G51Q)* and *MX2(T151A)* mutant vectors were generated by site-directed mutagenesis (QuikChange II XL, Agilent Technologies) according to the manufacturer's instructions. The herpesvirus-specific luciferase reporter plasmid pGL-T9G has previously been described[60].

**Viruses**. All viruses used in this study apart from KSHV were grown on A549 cells at 37 °C and 5% $CO_2$ and their infectious titres were determined on A549 cells (expressed as median tissue culture infective dose, $TCID_{50}$, as estimated by the Spearman–Kaerber method) unless specified otherwise. HSV-1 strain MacIntyre (VR-539) and strain F (VR-733) were purchased from ATCC. HSV-1 strain C12 has previously been described[48] and was kindly provided by S. Efstathiou (University of Cambridge, Cambridge, UK). HSV-2 strain G (ATCC, VR-734) was grown at 34.5 °C and 5% $CO_2$. IAV strain WSN (A/WSN/1933 (H1N1)) was grown and titrated by plaque assay on Madin–Darby canine kidney cells in the presence of 0.5 μg/ml tosyl phenylalanyl chloromethyl ketone (TPCK)-treated trypsin (Sigma-

Aldrich). HAdV-C5 (wt300) has previously been described[29]. Recombinant Kaposi's sarcoma-associated herpesvirus (rKSHV.219[51]) was obtained from BrK.219 cells[72] (generously provided by Thomas Schulz, Hannover Medical School, Hannover, Germany) treated with Roswell Park Memorial Institute (RPMI) 1640 medium (Life Technologies) containing 0.625 µg/ml anti-human IgM (µ-chain-specific, Southern Biotech), 0.05 µg/ml 12-O-tetradecanoylphorbol-13-acetate and 8% FBS. Cell-free supernatants were centrifuged at 30,000 × g and 4 °C for 2 h and titres of virus concentrates were determined by flow cytometric analysis of GFP-positive HEK-293T cells 48 h p.i. on a FACSCanto II (BD Biosciences). EdC genome-labelled HSV-1 was generated and purified as follows. A549 cells were infected with HSV-1 strain F at a low MOI and incubated until a complete cyto-pathic effect developed. EdC was added to the cells at 4 h p.i. to a final con-centration of 2.5 µM. EdC was synthesised as described[73]. Extracellular medium was harvested and the cellular debris was removed by low-speed centrifugation. The virus-containing supernatant was centrifuged at 27,500 × g and 4 °C for 90 min. The resulting pellets were resuspended by gentle shaking overnight at 4 °C in MNT buffer (30 mM 2-(N-morpholino)ethanesulfonic acid, 100 mM NaCl, 20 mM Tris-HCl, pH = 7.4). After waterbath sonication (three times for 30 s each at 4 °C), the suspension was layered on top of a discontinuous Nycodenz® gradient (20% and 30% Nycodenz®, SERVA Electrophoresis GmbH, Heidelberg) and centrifuged at 140,000 × g and 4 °C for 2 h. The virus-containing band accumulated at the interface was collected, aliquoted, snap frozen in liquid nitrogen and stored at −80 °C.

**RNA interference.** Transfection of siRNA was performed using Lipofectamine RNAiMAX reagent (Life Technologies) according to the manufacturer's standard reverse transfection protocol, unless stated otherwise. Briefly, transfection com-plexes were prepared using 30 nM siRNA and 1 µl transfection reagent in a 24-well format, unless stated otherwise. Cells in suspension were added to the complexes and incubated for 24 h at 37 °C. Transfection complexes were removed by washing with phosphate-buffered saline (PBS) and cells were incubated for another 24 h in growth medium. The siRNAs used in this study were purchased from Qiagen (siNT, custom siRNA, target sequence 5′-AAG CGT TCG TCC TAT GAT CGA-3′; siMxB UTR, Cat. No. SI00038206, target sequence 5′-CAG CAT TGT AAC TGC TTA ATA-3′; siMxB #1, Cat. No. SI00038220, target sequence 5′-TAG GCA TAC TTG CAA CTA ATA-3′; siRPS27A, custom siRNA, target sequence 5′-GCT GGA AGA TGG ACG TAC T-3′) and Dharmacon (siMxB #2, Cat. No. D-011736-01, target sequence 5′-GAG CAC GAU UGA AGA CAU A-3′).

**Indirect immunofluorescence assays.** Cells grown on glass coverslips were fixed with 3% (w/v) paraformaldehyde (PFA) in PBS for 10 min. Subsequent permea-bilisation and antibody staining was performed using PBS supplemented with 50 mM NH4Cl, 0.1% (w/v) saponin and 2% (w/v) bovine serum albumin (BSA, Roth AG, Arlesheim). Primary antibodies used were goat anti-MxB (Santa Cruz Bio-technology, sc-47197, 1:50), mouse anti-HSV-1/2 VP5 (Abcam, ab6508, 1:200), rabbit anti-HC polyclonal antibody (PAb) (directed against purified DNA-containing HSV-1 capsids, kindly donated by R. Eisenberg and G. Cohen, Uni-versity of Pennsylvania, Philadelphia, USA, 1:1000) and rat anti-LANA (Advanced Biotechnologies Inc., #13-210-100, 1:500). Secondary antibodies used were donkey anti-goat IgG coupled to Alexa Fluor 488, goat anti-mouse IgG-AF568, goat anti-rabbit IgG-AF568 and/or goat anti-rat IgG-AF555 (all from Life Technologies, 1:1000). DNA was counterstained using Hoechst 33342 (Life Technologies, 1:5000). Fluorescence images were recorded on a Leica TCS SP5 confocal laser scanning microscope (Leica Microsystems) maintained by the Center for Microscopy and Image Analysis, University of Zurich. To visualise the localisation of MxB in T98G cells, deconvolution was applied using Huygens version 17.04.

**IFN treatment prior to viral infection.** T98G cells were treated with 500 IU/ml recombinant human IFN-α2 (Roferon-A, Roche Pharmaceuticals) 18 h prior to virus infection. During and after virus inoculation, the IFN was omitted.

**Infection of T98G cells and epithelial cell lines.** Unless indicated otherwise, cells were inoculated with the indicated MOI in infection medium (DMEM, high glu-cose, supplemented with 0.2% (w/v) BSA, 20 mM 4-[2-hydroxyethyl]-1-piper-azineethanesulfonic acid (HEPES), 200 µM GlutaMAX, 100 IU/ml penicillin and 100 µg/ml streptomycin (Thermo Fisher Scientific)) for 30 min at room tempera-ture, washed with PBS, and then maintained in infection medium at 37 °C and 5% CO2 for the indicated time periods. For IAV, 0.5 µg/ml TPCK-treated trypsin (Sigma-Aldrich) was added to the infection medium after inoculation. For HAdV-C5, cells were inoculated at 37 °C and 5% CO2 for 120 min in 2% FBS medium (DMEM, high glucose, supplemented with 2% FBS, 200 µM GlutaMAX, 100 IU/ml penicillin and 100 µg/ml streptomycin (Thermo Fisher Scientific)). Cells were washed twice in 2% FBS medium and then incubated with 2% FBS medium at 37 °C and 5% CO2. HAdV-C5 was quantified from intracellular and extracellular virus obtained by scraping cells into the culture supernatant, snap freezing in liquid nitrogen and lysing with three consecutive freeze–thaw cycles. Lysates were cen-trifuged at 1000 × g for 5 min and titrated on fresh A549 cells. Number of infectious particles per ml was determined by fixing cells in 4% (w/v) PFA after 21 h, quenching with 25 mM NH4Cl, permeabilising with 0.5% Triton X-100 and

immunostaining with rabbit anti-protein VI antibody[74]. For HSV-1 strain C12, the infected cell cultures were maintained in 2% FBS medium and GFP fluorescence was monitored at 1–3 h intervals using the IncuCyte Zoom Live Cell Analysis System (Essen Bioscience). For HSV-2, inoculation was performed in 2% FBS medium for 60 min at 34.5 °C and 5% CO2. Cells were then washed with PBS and incubated with 2% FBS medium at 34.5 °C and 5% CO2. For IAV, VSV, HSV-1 and HSV-2 virus titre determination, supernatants from infected cell cultures were centrifuged at 1500 × g and 4 °C for 10 min and titrated on fresh A549 cells using TCID50 assay. For KSHV, cells were spinoculated for 1 h at 800 × g and 4 °C in growth medium and then shifted to 37 °C and 5% CO2. At 48 h p.i., cells were fixed with 1% (w/v) PFA and GFP expression was measured on an LSR II Fortessa flow cytometer (BD Biosciences) and analysed using FlowJo version 10.2.

**Sodium dodecyl sulphate-polyacrylamide gel electrophoresis and immunoblot assays.** Cells were washed in ice-cold PBS and lysed with radio-immunoprecipitation buffer (20 mM Tris-HCl, pH = 7.4, 100 mM NaCl, 1% NP-40, 0.1% (w/v) sodium dodecyl sulphate, 1% (w/v) sodium deoxycholate, 50 mM NaF, 50 mM β-glycerophosphate, 1 mM Na3VO4, 1 mM ethylenediaminete-traacetic acid, 1 mM dithiothreitol and cOmplete™ Protease Inhibitor Cocktail (Roche Applied Science)) for 10 min on ice. Lysates were cleared using QIAsh-redder spin columns (Qiagen), boiled in Laemmli buffer and resolved on NuPAGE™ 4–12% polyacrylamide gels (Invitrogen). Proteins were transferred onto a nitrocellulose membrane and stained using the following antibodies: goat anti-MxB (Santa Cruz Biotechnology, sc-47197, 1:500), rabbit anti-MxA (Novus Bio-logicals, H00004599-D01P, 1:1000), mouse anti-HSV-1/2 VP16 monoclonal anti-body LP1[75] (kindly donated by A. Minson and H. Browne, University of Cambridge, Cambridge, UK, 1:5000), rabbit anti-GAPDH (Santa Cruz Bio-technology, sc-25778, 1:3000) and mouse anti-GST (Santa Cruz Biotechnology, sc-57753, 1:1000). Densitometric analyses were performed using MultiGauge version 3.0. Values were normalised to the loading control and are represented relative to the appropriate control condition. Uncropped scans of all immunoblots can be found in Supplementary Figs. 8–12.

**RNA extraction and HSV-1 RT-qPCR.** Total RNA from infected A549-GST and A549-MxB cells was obtained by lysis in TRIzol® Reagent (Invitrogen) and RNA extraction using RNeasy Mini Kit (Qiagen) according to the manufacturer's instructions. DNA was digested with DNase I (Thermo Fisher Scientific), and cDNA synthesis was performed using random primers (Promega) and SuperScript III First-Strand Synthesis System for RT-PCR (Invitrogen). cDNA from viral transcripts was quantified by qPCR using GAPDH gene as endogenous control and primers specific for the RL2, RS1 and UL29 genes using SYBR green (Life Tech-nologies). HSV-1 cDNA sequences were amplified at the following conditions: 3 min at 95 °C, 39× (15 s at 95 °C, 60 s at 45 °C), 10 s at 95 °C, 5 s at 65 °C, 5 s at 95 °C. The following primers were used (in 5′–3′ orientation): RL2-forward: CCT GTC GCC TTA CGT GAA CA and RL2-reverse: ACA CGG ATT GGC TGG TGT AG; RS1-forward: GTG AGA CCC GAA GAC GCA AT and RS1-reverse: CAT CTC TAC CTC AGT GCC GC; UL29-forward: TGG CTT TTC GGA CTA CAC CC and UL29-reverse: TTC GAA GGC CGT GAA CGT AA; GAPDH-forward: GAA GGT GAA GGT CGG AGT C and GAPDH-reverse: GAA GAT GGT GAT GGG ATT TC. Ct values were normalised to GAPDH (ΔCt) and relative virus mRNA levels were calculated using the $2^{-\Delta\Delta Ct}$ method.

**Assays for HSV-1 protein expression and HSV-1 VP5 exposure.** A549-GST and A549-MxB cells were inoculated with HSV-1 strain F at the indicated MOI in 2% FBS medium for 60 min on ice in the presence or absence of 100 µg/ml cyclo-heximide (Sigma-Aldrich). The inoculum was removed and replaced with 2% FBS medium with or without 100 µg/ml cycloheximide. Cells were incubated at 37 °C for the indicated time periods, washed twice in PBS and fixed with 3% (w/v) PFA in PBS for 10 min. Fixed samples were permeabilised with 0.5% Triton X-100 in PBS for 5 min, washed three times, and then blocked with 5% goat serum for 30 min at room temperature. HSV-1 major capsid protein was stained using mouse anti-HSV-1/2 VP5 antibody (Abcam, ab6508, 1:200). For staining of HSV-1 immediate-early and early proteins, rabbit anti-ICP0 PAb (kindly provided by Ben Hale, 1:100), mouse anti-ICP4 (Abcam, ab6514, 1:2500) and mouse anti-ICP8 (Abcam, ab20193, 1:100) were used. Secondary antibodies used were goat anti-rabbit IgG-AF488 or goat anti-mouse IgG-AF488 (Life Technologies, 1:1000). DNA was counterstained using Hoechst 33342 (Life Technologies, 1:5000). Fluorescence images were recorded on a Leica TCS SP5 confocal laser scanning microscope (Leica Microsystems) maintained by the Center for Microscopy and Image Analysis, University of Zurich. For quantification of VP5 exposure, a region of interest (ROI) along the central focal plane around each cell nucleus (defined by the DNA counterstain and subsequently enlarged to cover the cytoplasm around it) was determined and VP5-positive foci within each ROI were quantified. For quantification of ICP0, ICP4, ICP8 and VP5 de novo protein expression at early time points of infection, the mean fluorescence intensity in the nucleus (defined by the DNA counterstain) was determined. All data were quantified using custom-made ImageJ version 1.48 macros.

**Analysis of HSV-1 capsid and genome localisation**. A549-GST and A549-MxB cells were inoculated with EdC genome-labelled HSV-1 strain F for 1 h on ice with constant rocking in RPMI 1640 medium (Life Technologies) containing 0.2% (w/v) BSA and 20 mM HEPES to allow virus binding. After this period, cells were washed twice with fresh medium and either fixed immediately in 4% (w/v) PFA in PBS or fixed at intervals after incubation at 37 °C and 5% $CO_2$ in the presence of 100 μg/ml cycloheximide (Sigma-Aldrich) to prevent synthesis of viral proteins. For detection of incoming capsids, cells were stained with rabbit anti-HC PAb. After immunostaining, viral genomes were visualised using Copper(I)-catalysed azide alkyne cycloaddition (click) as previously described[28]. Briefly, samples were stained for 2 h at room temperature with freshly prepared click chemistry staining mix containing 10 μM AF488-azide (Life Technologies), 1 mM $CuSO_4$ (Sigma-Aldrich), 10 mM sodium ascorbate (Sigma-Aldrich), 10 mM amino-guanidine (Sigma-Aldrich) and 1 mM Tris(hydroxypropyltriazolyl)methylamine (Sigma-Aldrich). Samples were stained with 4′,6-diamidino-2-phenylindole (DAPI, Life Technologies) for cell nuclei and succinimidyl ester-AF647 (Life Technologies) for cell outlines, and embedded in DAKO medium (Dako Schweiz AG, Baar). Fluorescence images were recorded on a Leica TCS SP8 confocal laser scanning microscope (Leica Microsystems) maintained by the Center for Microscopy and Image Analysis, University of Zurich. Images were analysed using a custom-made CellProfiler version 2.1.1 pipeline and KNIME version 2.12.2.

**Transmission electron microscopy**. A549-MxB cells pretreated with siRNA were spinoculated with HSV-1 strain MacIntyre (MOI = 500, virus stock produced in 2% FBS medium and supplemented with 20 mM HEPES prior to inoculation) at $800 \times g$ and 4 °C for 60 min in the presence of 100 μg/ml cycloheximide (Sigma-Aldrich). The inoculum was removed and cells were incubated in infection medium in the presence of 100 μg/ml cycloheximide for 3 h. Samples were processed for TEM and analysed by TEM on a Zeiss EM 902A with modifications of previously published protocols[76]. Prior to fixation, cells were washed twice with warm PBS containing 1 mM $CaCl_2$ and 0.5 mM $mgCl_2$ (PBS + Ca/Mg). Cells were fixed in 2% glutaraldehyde in PBS + Ca/Mg for 30 min at room temperature. Preparation of samples for TEM was performed as follows. Post-fixation: short wash in PBS, 3× wash in $H_2O$ for 5 min, incubation with 1% osmium tetroxide + 1.5% potassium ferricyanide in $H_2O$ at 4 °C for 1 h, 3× wash in PBS for 3 min, 2× wash in $H_2O$ for 5 min and incubation with 2% uranyl acetate in $H_2O$ overnight at 4 °C. Sample dehydration: 30% acetone for 5 min, 50% acetone for 5 min, 70% acetone for 30 min, 90% acetone for 10 min, 100% acetone for 5 min and 100% acetone for 10 min. Sample embedding: 48% epoxy resin, 16% dodecenyl succinic anhydride, 34% methyl nadic anhydride and 2% benzyldimethylamine. Resin was allowed to polymerise at 60 °C for 2 days. Sample mounting, staining and imaging: ultra-thin sections (85 nm) obtained with a Leica Ultracut UCT ultramicrotome (Leica Microsystems) were mounted on copper grids with parlodion-carbon support film, placed sideways on a droplet of 2% uranyl acetate in $H_2O$ for 30 min, immersed repeatedly in $H_2O$, air dried overnight, placed sideways on a droplet of Sato's lead staining solution for 20 min, immersed repeatedly in $H_2O$, air dried for 30 min and imaged at 80 kV and ×50,000 magnification.

**Herpesvirus luciferase reporter assay**. Transient transfections were performed using jetPRIME® reagent (Polyplus Transfection) according to the manufacturer's instructions. Transfection complexes were prepared at a DNA:transfection reagent ratio of 1:2 with a total amount of 500 ng DNA per well (24-well plate assay format). Specifically, 200 ng pGL-T9G and increasing amounts of expression plasmids were used (37.5, 75 and 150 ng). The total amount of DNA per well was adjusted using the empty expression plasmid. After the infection period, luciferase activity was determined using the BrightGlo™ Luciferase Assay System (Promega) on an Envision 2104 plate reader (Perkin Elmer).

**Transient overexpression for analysis of MxB localisation**. Transient transfections for immunofluorescence analysis of MxB variants were performed using ViaFect™ reagent (Promega) according to the manufacturer's instructions. Transfection complexes were prepared at a DNA:transfection reagent ratio of 1:3 with a total amount of 500 ng expression plasmid per well (24-well plate assay format).

**Statistical analyses**. Statistical analyses were performed using R version 3.4.1[77]. Individual statistical tests are specified within the figure legends. Count data with more than two groups were analysed by Kruskal–Wallis test and pairwise Wilcoxon's rank sum tests. All other grouped data were analysed by analysis of variance (ANOVA) under consideration of the assumptions of normality and homogeneity of variance[78]. Multiple comparison tests with user-defined contrasts were performed with the glht function in the multcomp package. For non-orthogonal multiple comparisons, α-levels were corrected according to Holm's method[79].

**Code availability**. Computer code used for immunofluorescence image quantification is available on https://github.com/greberlab/2018_NatComm_MxB.

**Data availability**. The authors declare that the data supporting the findings of this study are available within the article and its Supplementary Information files, or are available on request.

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

## Acknowledgements

We thank our colleagues Silke Stertz, Ben Hale and Yohei Yamauchi (University of Bristol, Bristol, UK) for discussions, Eva Moritz, Ruth Hefti, Bernd Vogt and Elisabeth Schraner for technical assistance, Roselyn Eisenberg and Gary Cohen (University of Pennsylvania, Philadelphia, USA) for their kind gift of antibodies and advice, Beate Sodeik (Hannover Medical School, Hannover, Germany) for discussions, Alexandra Trkola for the pNLluc-AM vector, Naoki Inoue (Gifu Pharmaceutical University, Gifu, Japan) for the pGL-T9G reporter plasmid, Mirco Ponzoni (IRCCS Istituto G. Gaslini, Genova, Italy) for providing HUVECs, Simon Crameri (Institute of Integrative Biology, ETH Zurich, Zurich, Switzerland) for help with statistical analyses and Abhilash Kannan for identifying miRNA-homologous sequences in the siRNAs. This work was supported by the Canton of Zurich, Switzerland. KSHV research in the laboratory of C.M. is supported by Cancer Research Switzerland (KFS-4091-02-2017).

## Author contributions

M.C. and J.P. designed the study and wrote the manuscript with inputs from U.F.G. and other authors. M.C. and R.W. established cell lines, performed immunofluorescence assays, immunoblot assays, infection assays, titrations and RNA interference assays. M.C. performed the HSV-1 VP5 exposure assay, luciferase assays and cell viability assays. M.B.

carried out the adenovirus experiments and the experiment regarding the localisation of HSV-1 capsids and genomes during virus entry. N.C., M.C. and C.G. carried out the experiments related to KSHV. F.S. and T.K. were involved in the HSV titrations. M.C., F. D.F. and A.Z. designed and carried out the HSV-1 RNA and DNA analyses. K.B., U.F.G. and M.C. performed and analysed the TEM experiment. C.M., C.F. and U.F.G. were involved in the study design and provided material.

## Additional information

**Competing interests:** The authors declare no competing interests.

