## [Peer Review File · Nature Communications]

Reviewers' comments:

Reviewer #1 (Remarks to the Author):

Crameri et al. claim that human myxovirus resistance protein B (MxB) is a potent human herpesvirus restriction factor when over expressed in human established cell lines. They suggest that MxB restricts early steps of the HSV-1 life cycle after tegument dissociation and transport of the capsid to the nucleus but before transcription of viral immediate-early genes. In addition, they propose that MxB activity against herpesviruses is dependent upon its GTPase activity. Unfortunately, based upon numerous concerns, including the assays employed, quality of data, minor differences detected, technical issues (including the use of a single cloned cell lines overexpressing MxB), and confounding data in figures 3 and 4, their results do not necessarily support their interpretation from a biologically meaningful perspective.

MAJOR CONCERNS

1) Line 98 and Figure 1. The authors claim that interferon-induced MxB inhibits HSV-1 replication. The results of the plaque assay, which is the gold standard for measuring productive viral replication, are remarkably unimpressive. The effects with the McIntyre strain at best approach a 6-fold increase when MxB is depleted in interferon-treated cultures. The effects using strain F are even less impressive. Even if these differences were numerically statistical (no statistical analysis is provided), the biological relevance of these marginal differences in infectious virus production is questionable at best.

2) According to the methods section, it appears that clones of A549 cells expressing either GST or MxB were isolated and a single clone was selected and used for these experiments. Results need to be shown with multiple clones that are proven to be independent or by forgoing the use of clones altogether and using a pool resulting from a high efficiency transfection to rule out the possibility that the results stem from variants within the population.

3) The relevance of A549 cells to KSHV biology is questionable. Also, the authors claim to be using rKSHV.219, which has an RFP lytic reporter and a GFP latent infection reporter. Besides inadequately explaining this, the authors assume the small approximately 3-fold increase in GFP-positive cells detected following MxB depletion in MxB-overexpressing cells results from restriction of virus replication. As GFP reports on latent infection, other explanations of this are certainly possible.

4) Line 168: The statement, "MxB blocks HSV-1 infection at a level upstream of immediate-early gene expression" is not necessarily true based upon the data presented. The data fails to exclude the possible impact of post-transcriptional events, thus the conclusion that MxB blocks upstream of IE gene expression is not supported by data. The assays used here measure steady-state accumulation of mRNA and protein and as such, other interpretations are possible.

5) Figure 3B shows at best modest differences in variation of nuclear protein accumulation at 4.5h post-infection. This is especially true for IE proteins ICP4 and 0. Should the infection progress more slowly in response to MxB overexpression, it is not surprising that greater reduction of early (ICP8) and late (VP5) proteins be observed in MxB compared to GST expressing cells. A similar argument can be applied to the reported HSV-1 F mRNA levels in Fig. 3A,. Why the C12 recombinant behaves differently than an accepted WT strain (strain F) is not addressed.

6) Line 195 and figure 4: these effects, while numerically significant, are miniscule (ie at best a 3-fold reduction of cytoplasmic full capsids upon MxB depletion with a single siRNA). A better, more biological readout of genome delivery into nuclei would be to assay IE transcript production in response to VP16. If the MxB block were crucial, one would expect to see a substantial reduction of IE mRNA (which is directly responsive to VP16) . Unfortunately, data in figure 3 shows only minor effects on IE mRNA accumulation (ICP0, ICP4), suggesting in fact that most capsids deliver genome payloads (at a reasonable MOI of 0.5) into the nucleus where IE transcription is stimulated by VP16. So it appears figure 3a contradicts what is presented in figure 4. If the genomes are not being delivered into nuclei, how would one detect IE gene transcription? It looks as if an excessive MOI of 500 is required to see these minor defects in loaded capsid accumulation in Mx B expressing cells, calling into question whether this is an artifact of MOI.

Reviewer #2 (Remarks to the Author):

The manuscript by Crameri et al reports that the interferon inducible MX2 gene product, MxB, is a novel restriction factor for human alpha- and gamma-herpesviruses.

It is clear that a more complete understanding of intrinsic anti-viral responses to herpesvirus infections is crucial to understanding virus/host interactions and, ultimately, viral pathology of these persistent human pathogens.

In general, the story presented is novel and timely and does expand our knowledge of the players involved in intrinsic anti-herpesvirus responses but there are a number of points that the authors need to address:

1) Figure 1b and c – statistical significances should be shown.

2) Figure 2 d and f. I am uncertain what the authors mean in line 262 regarding the data shown in these figures with respect to different titres of viruses (MacIntyre was about 10,000 fold lower than F). Weren't the same titres of MOI of 0.5 used for both analyses? That said, this difference between MacIntyre and F might be explained by differences in levels of defective viruses. For instance what was the particle/genome to PFU ratio of these different virus stocks – did they both express equivalent levels of e.g. ICP0 for equivalent particles/genomes. It is possible that one virus had more defective particles in them, delivering large numbers of particles/genomes without equivalent levels of expression of viral inhibitors of anti-viral responses.

Also, statistical significances should be shown for e.g. figure 1d-f.

3) Figure 4c. The purity of the fractionation of nuclei is crucial for this analysis. I accept that supplementary figure 4 goes some way to this but genomic DNA PCRs should also be

shown.

4) Figure 5. The pGL-T9G reporter system, originally using a stably transfected ORF9 luciferase reporter MeWo cell line, is elegant and has been validated for quantifying VZV infection and HSV-1 infection but I have not seen this system based on transient transfection of the reporter in HeLa cells. Has this been validated to show strict correlation between HSV infection and luciferase reporter expression? It would be much better to recapitulate the effect or lack of effect of these MxB mutants by stable transfection in e.g. A459 cells followed by superinfection with HSV-1.

Crameri *et al.* - Reviewers' comments

REVIEWER #1 (Remarks to the Author):

Crameri *et al.* claim that human myxovirus resistance protein B (MxB) is a potent human herpesvirus restriction factor when over expressed in human established cell lines. They suggest that MxB restricts early steps of the HSV-1 life cycle after tegument dissociation and transport of the capsid to the nucleus but before transcription of viral immediate-early genes. In addition, they propose that MxB activity against herpesviruses is dependent upon its GTPase activity. Unfortunately, based upon numerous concerns, including the assays employed, quality of data, minor differences detected, technical issues (including the use of a single cloned cell lines overexpressing MxB), and confounding data in figures 3 and 4, their results do not necessarily support their interpretation from a biologically meaningful perspective.

MAJOR CONCERNS

REVIEWER #1, COMMENT 1:

1.1) Line 98 and Figure 1: The authors claim that interferon-induced MxB inhibits HSV-1 replication. The results of the plaque assay, which is the gold standard for measuring productive viral replication, are remarkably unimpressive. The effects with the McIntyre strain at best approach a 6-fold increase when MxB is depleted in interferon-treated cultures. The effects using strain F are even less impressive.

In our experiments for Fig. 1b, we refined the methods by pretreating T98G cells with less exogenous interferon (IFN), infecting with a lower MOI of HSV-1, and allowing 32 hours of infection with both HSV-1 strains. This procedure resulted in approximately 1,000-fold reduction of HSV-1 titres after IFN pretreatment (see Fig. 1b). In order to account for non-specific effects of siRNA transfection, relative titres are shown. In this setting, depletion of a single ISG, *MX2*, using two independent *MX2*-specific siRNAs increased viral titres 7.5-25-fold (see Fig. 1b). The observed effects are comparable to the potency of endogenous MxB against HIV-1 (see Goujon *et al.* 2013¹, Figure 2a).

1.2) Even if these differences were numerically statistical (no statistical analysis is provided), the biological relevance of these marginal differences in infectious virus production is questionable at best.

For each experiment, we performed ANOVA statistical tests², followed by multiple comparisons between groups with *post-hoc* correction, demonstrating that the contribution of endogenous MxB in IFN-mediated restriction of three distinct HSV-1 strains is statistically highly significant (see Fig. 1b, c). For a detailed description of our statistical analysis, see also our response to Reviewer #2, comment 1.

Type I IFN induces the expression of several hundred IFN-regulated genes (ISGs), many of which have intrinsic antiviral activity³. Some of these ISGs, in particular PKR, have been reported to exert antiviral activity against HSV-1^{4,5}. The biological relevance of MxB in the context of IFN is demonstrated with the PKR-sensitive strain C12 that exhibits a rescue phenotype upon *MX2* silencing that is at least as pronounced as the rescue phenotype observed with *PKR* silencing (compare Fig. 1c with Supplementary Fig. 1c).

REVIEWER #1, COMMENT 2:

2) According to the methods section, it appears that clones of A549 cells expressing either GST or MxB were isolated and a single clone was selected and used for these experiments. Results need to be shown with multiple clones that are proven to be independent or by forgoing the use of clones altogether and using a pool resulting from a high efficiency transfection to rule out the possibility that the results stem from variants within the population.

Initially we tested pools of A549 cells following transduction and selection. However, in these pools of transduced cells we observed only 5-10% of stably MxB expressing cells, making them unsuitable for titration experiments.

As suggested by the reviewer, we generated several A549-MxB clones from independently transduced A549 pools and tested their capacity to inhibit HSV-1 and HSV-2. These clones exhibited the same anti-herpesvirus phenotype as the original A549-MxB clone (compare the three lower panels in Supplementary Fig. 2a and compare Fig. 2d with Supplementary Fig. 2b).

We agree with the reviewer that selected clones may exhibit phenotypes that are not solely dependent on MxB expression. Nevertheless, restriction of the different herpesviruses in three independent cell clones was only relieved when ectopically expressed MxB was depleted using siRNAs directed against the coding sequence of *MX2*, showing the strict relationship between MxB expression and antiviral activity in several cellular backgrounds (compare Fig. 2d with Supplementary Fig. 7b as well as Supplementary Fig. 2a, b with Supplementary Fig. 7e).

REVIEWER #1, COMMENT 3:

3.1) *The relevance of A549 cells to KSHV biology is questionable.*

We strongly believe that the human epithelial A549 cell line represents a valid model for natural KSHV infection. Oropharyngeal epithelial cells were found to be a source of natural KSHV infection in immunocompetent donors, and it is likely that KSHV replicates in the oral cavity and is shed into the saliva⁶. It has also been suggested that oral transmission of KSHV occurs between healthy individuals^{7,8}.

KSHV causes malignancies in immunocompromised individuals, including Kaposi's sarcoma, and the virus is found primarily in endothelial cells within these tumour tissues⁹. Hence, we also tested MxB-dependent restriction of KSHV in primary human umbilical vein endothelial cells (HUVECs), as they are often used to study KSHV biology¹⁰. Unfortunately, transfection or transduction of HUVECs resulted in massive production of type I IFN and consequently expression of many ISGs, including MxB. Assessment of the antiviral effect of endogenous MxB was not feasible under these conditions. Transfection of siRNA directed against *IRF9*, a transcription factor required for type I IFN-dependent ISG expression¹¹, reduced but did not abrogate accumulation of endogenous MxB and other ISGs. Nevertheless, using this strategy we were able to assess restriction of KSHV by ectopically expressed MxB. Although clearly less pronounced compared to A549 cells, ectopic MxB restricted KSHV replication in HUVECs and this restriction could be completely relieved by co-transfecting siRNA directed against *MX2* (see Supplementary Fig. 3b, c). KSHV lytic infection, as measured by RFP fluorescence, was detected in HUVECs with our system, but it remained only marginally discernible from background fluorescence, thus precluding further analysis.

3.2) Also, the authors claim to be using rKSHV.219, which has an RFP lytic reporter and a GFP latent infection reporter. Besides inadequately explaining this, the authors assume the small approximately 3-fold increase in GFP-positive cells detected following MxB depletion in MxB-overexpressing cells results from restriction of virus replication. As GFP reports on latent infection, other explanations of this are certainly possible.

In order to provide the reader with the relevant information about the recombinant KSHV used in our study, we added a detailed explanation of its characteristics in the *Results* section (see lines 164-166 & lines 183-187). In particular, we demonstrate that GFP reporter expression and endogenous latency-associated nuclear antigen (LANA) protein expression do indeed correlate (see Supplementary Fig. 3a, bottom right panel), as was suggested in a previous study¹².

Immunostaining of LANA at 16 hours after inoculation, as well as GFP reporter expression, revealed that recombinant KSHV was able to quickly establish latent infection in A549 cells (see Fig. 2f and Supplementary Fig. 3a). On the other hand, lytic gene expression as measured by RFP fluorescence was not detectable in KSHV-infected A549 cells in the first 2 days after inoculation, which is consistent with previous reports that KSHV primary infection in epithelial cells and tissues does not induce the lytic cycle spontaneously but rather follows a latency program in various systems *in vitro* and *in vivo*^{6,12,13}.

As the reviewer pointed out, the GFP reporter (in absence of concomitant RFP expression) is a measure of latent KSHV infection. Against this background, the increase of GFP-positive cells in *MX2*-depleted A549 cell populations as compared to *MX2*-overexpressing cell populations (see Fig. 2e) represents a direct effect of MxB in a single cycle of infection, most likely at the level of KSHV entry into the nucleus.

REVIEWER #1, COMMENT 4:

4) Line 168: The statement "MxB blocks HSV-1 infection at a level upstream of immediate-early gene expression" is not necessarily true based upon the data presented. The data fails to exclude the possible impact of post-transcriptional events, thus the conclusion that MxB blocks upstream of IE gene expression is not supported by data. The assays used here measure steady-state accumulation of mRNA and protein and as such, other interpretations are possible.

We acknowledge the possibility that, solely based on the data provided in Fig. 3, we cannot exclude inhibitory effects of MxB on a transcriptional or post-transcriptional level. The title of the paragraph describing our results shown in Fig. 3 has thus been changed to "MxB inhibits accumulation of HSV 1 immediate-early transcripts and proteins" (see line 200).

However, in Fig. 4 and Supplementary Fig. 4, we show that the MxB-dependent inhibition HSV-1 occurred at an earlier step of the HSV-1 life cycle. Using four different methods (immunofluorescence analysis, click chemistry, TEM, and qPCR) we observed that HSV-1 genome uncoating and translocation to the nucleus was inhibited in A549-MxB cells. These findings strongly suggest that MxB-mediated restriction of HSV-1 occurs at a step before immediate-early (IE) gene expression (see Fig. 4b, c and Supplementary Fig. 4a). Even so, we do not exclude that MxB blocks HSV-1 infection also at later stages in the life cycle (see lines 375-378 in the *Discussion* section).

REVIEWER #1, COMMENT 5:

5.1) Figure 3B shows at best modest differences in variation of nuclear protein accumulation at 4.5h post-infection. This is especially true for IE proteins ICP4 and 0. Should the infection progress more slowly in response to MxB overexpression, it is not surprising that greater reduction of early (ICP8) and late (VP5) proteins be observed in MxB compared to GST expressing cells. A similar argument can be applied to the reported HSV-1 F mRNA levels in Fig. 3A.

The experiments shown in Fig. 3 were designed to assess the activity of MxB early in the first cycle of infection to avoid indirect amplification effects due to multicycle growth. We measured viral mRNA and protein expression at the earliest possible time points post infection. Hence, differences in viral mRNA and/or protein accumulation between A549-GST and A549-MxB cells are not expected to be of great magnitude.

We extended our analysis of HSV-1 mRNA and protein accumulation at early time points of infection by including data of HSV-1 strain MacIntyre (see Fig. 3a, b). Our results with the strains MacIntyre and F are similar with regards to mRNA and protein levels, demonstrating the reproducibility of our findings.

We agree with the reviewer that the effect of MxB on HSV-1 late protein (e.g. VP5) expression is likely a consequence of impairments in IE gene expression. We clarified this point in the *Results* section (see lines 212-216) and in the *Discussion* section (see lines 360-364).

5.2) Why the C12 recombinant behaves differently than an accepted WT strain (strain F) is not addressed.

We agree with the reviewer that the differences in HSV-1 mRNA accumulation between A549-GST and A549-MxB cells were more pronounced in the C12 strain when compared to strains MacIntyre and F. This observation could either result from altered kinetics in life cycle progression, enhanced sensitivity to MxB, or a combination of both. As a consequence, the impact of MxB on C12 mRNA accumulation may appear higher at the measured time point 4.5 h post infection. We added a clarifying statement in the *Results* section (lines 209-211).

REVIEWER #1, COMMENT 6:

6.1) Line 195 and figure 4: These effects, while numerically significant, are miniscule (i.e. at best a 3-fold reduction of cytoplasmic full capsids upon MxB depletion with a single siRNA). A better, more biological readout of genome delivery into nuclei would be to assay IE transcript production in response to VP16. If the MxB block were crucial, one would expect to see a substantial reduction of IE mRNA (which is directly responsive to VP16). Unfortunately, data in figure 3 shows only minor effects on IE mRNA accumulation (ICP0, ICP4), suggesting in fact that most capsids deliver genome payloads (at a reasonable MOI of 0.5) into the nucleus where IE transcription is stimulated by VP16.

Our transmission electron microscopy (TEM) data (see Fig. 4c) cannot distinguish capsids delivering genomes to the nucleoplasm from capsids misdelivering genomes to e.g. the cytoplasm. However, genome misdelivery is a common phenomenon and has been observed in a significant fraction of incoming viral capsids¹⁴⁻¹⁷. It is therefore not necessarily expected that a high degree of HSV-1 genome uncoating translates into efficient initiation of IE gene transcription in the light of efficient VP16 transport to the nucleus. At an MOI of 0.5 as was used in the qRT-PCR assay shown in Fig. 3a, HSV-1 genome delivery to the nucleus may have been successful in only a small fraction of cells. This may explain the observed degree of reduction in IE mRNA accumulation in A549-MxB cells. Even so, we would like to point out that our qRT-PCR data are statistically significant and can be reproduced with three distinct HSV-1 strains.

6.2) So it appears figure 3a contradicts what is presented in figure 4. If the genomes are not being delivered into nuclei, how would one detect IE gene transcription?

In addition to our TEM data, where we evaluated DNA uncoating irrespective of genome misdelivery (see Fig. 4c), and qPCR data, where we determined the accumulation of HSV-1 genomic DNA in the nucleus (see Supplementary Fig. 4a), we now provide new data using a click chemistry approach¹⁶ that allows direct detection of individual HSV-1 genomes delivered to the nucleus (see Fig. 4b, middle panel). These data are strongly in line with our TEM and qPCR data, indicating that the activity of MxB indeed results in a reduced accumulation of incoming viral genomes in the nucleus.

There is no contradiction between the data presented in Figs. 3a and 4. Firstly, we observed a significant reduction of HSV-1 genome translocation in A549-MxB cells rather than a complete block (see Fig. 4b, middle panel). The genome foci number in A549-MxB cells slightly increased between 0 min and 120 min post infection ($p = 0.0082$ as determined by pairwise Wilcoxon rank sum test), indicating that nuclear translocation of HSV-1 genomes did occur in a subpopulation of cells expressing MxB. Secondly, viral genomic DNA in nuclear fractions of infected A549 cells expressing MxB remained on a detectable level (see Supplementary Fig. 4a), explaining why we measured some IE transcripts in A549-MxB cells (see Fig. 3a).

In sum, all our findings support a model where MxB inhibits HSV-1 capsid trafficking and/or genome uncoating (see Figs. 4b, left panel, 4c, and Supplementary Fig. 4a), but not capsid uncoating from the tegument (see Fig. 4a). This results in less efficient delivery of HSV-1 genomes to the nucleus (see Fig. 4b, middle panel), and, as a consequence, diminished IE gene transcription.

6.3) It looks as if an excessive MOI of 500 is required to see these minor defects in loaded capsid accumulation in MxB expressing cells, calling into question whether this is an artifact of MOI.

TEM analyzes ultra-thin sections (<100 nm) with subcapsid size resolution in the nanometer range. We used sections of 85 nm thickness, which encompass about one hundredth of the thickness of a confluent A549 cell monolayer (approx. 5-10 μm , own observation). On top of that, our analysis of HSV-1 genome uncoating included only virus capsids that were (i) clearly distinguishable from surrounding structures and (ii) embedded in a centered manner, to differentiate between DNA-containing capsids and empty capsids. Evidently, such viral structures would be extremely unlikely to be observed in low-MOI settings. The use of HSV-1 at an MOI of 500 is an established procedure for TEM analysis and is required to obtain a statistically meaningful sample¹⁸. Considering the ability of a single cell to produce thousands of infectious virus particles within a short time¹⁹, likely to infect neighbouring cells simultaneously, an MOI of 500 may well reflect a physiological condition on a single-cell level.

REVIEWER #2 (Remarks to the Author):

The manuscript by Crameri et al reports that the interferon inducible MX2 gene product, MxB, is a novel restriction factor for human alpha- and gamma-herpesviruses.

It is clear that a more complete understanding of intrinsic anti-viral responses to herpesvirus infections is crucial to understanding virus/host interactions and, ultimately, viral pathology of these persistent human pathogens.

In general, the story presented is novel and timely and does expand our knowledge of the players involved in intrinsic anti-herpesvirus responses but there are a number of points that the authors need to address:

REVIEWER #2, COMMENT 1:

1) *Figure 1b and c – statistical significances should be shown.*

We now analysed the grouped data for each virus strain using an ANOVA model with two explanatory variables (interferon treatment and siRNA treatment). The ANOVA model was adapted to meet the assumptions of normality and homogeneity of variance². An interaction term was also included, showing that siRNA treatment has an antagonising effect on the effect of interferon treatment, as expected. In addition, multiple comparisons between groups with *post-hoc* correction are shown, demonstrating high statistical significance.

REVIEWER #2, COMMENT 2:

2.1) *Figure 2 d and f. I am uncertain what the authors mean in line 262 regarding the data shown in these figures with respect to different titres of viruses (MacIntyre was about 10,000 fold lower than F). Weren't the same titres of MOI of 0.5 used for both analyses?*

Our wording in this section may not have been clear in the original manuscript. The factor 10,000 was referring to the inhibition of HSV-1 strain MacIntyre replication in A549-MxB cells as compared to A549-GST cells. However, as we point out in the next paragraph, this factor may have been overestimated due to inequalities of our virus inocula. Accordingly, we removed lines 262-271 from the original manuscript.

2.2) That said, this difference between MacIntyre and F might be explained by differences in levels of defective viruses. For instance what was the particle/genome to PFU ratio of these different virus stocks – did they both express equivalent levels of e.g. ICP0 for equivalent particles/genomes. It is possible that one virus had more defective particles in them, delivering large numbers of particles/genomes without equivalent levels of expression of viral inhibitors of anti-viral responses.

We agree with the reviewer that the number of physical particles may have differed among the virus stocks used, possibly influencing the interpretation of our results. Therefore we analysed the original stocks of the three different HSV-1 strains (MacIntyre, F, or C12) by electron microscopy to determine the particle numbers and compared them to the corresponding infectious virus titre. This analysis revealed that the ratio of physical particles to infectious particles was 4.5 times higher in the F strain compared to the MacIntyre strain. The F strain may therefore have delivered more factors antagonising the antiviral response, such as ICP0. We repeated the infection assays for Fig. 2d with HSV-1 stocks grown under identical conditions and observed that the pronounced differences between the two strains observed initially were no longer present. After replication of the infection assays three times, we are confident that the 75-fold inhibition in our A549-MxB cells compared to A549-GST cells accurately represents the situation in both strains.

2.3) Also, statistical significances should be shown for e.g. figure 2d-f.

In analogy to Fig. 1b, we performed ANOVA statistical tests for Fig. 2d-f with cell line and siRNA treatment as explanatory variables followed by multiple comparisons with *post-hoc* correction, demonstrating the high statistical significance of our results.

REVIEWER #2, COMMENT 3:

3) Figure 4c. The purity of the fractionation of nuclei is crucial for this analysis. I accept that supplementary figure 4 goes some way to this but genomic DNA PCRs should also be shown.

In order to address this concern, we generated genomic DNA qPCR data from cytoplasmic fractions as well as nuclear fractions using primers specific to the host nuclear gene *GAPDH* (see Supplementary Fig. 4b, right panels). For each individual sample, we observed virtually no leakage of nuclear material from the nuclear fractions to the cytoplasmic fractions during lysis. Specifically, cytoplasmic fractions contained at least 150 times less *GAPDH* DNA as calculated by the ΔC_t method. Contamination of nuclear fractions with cytoplasmic (viral) genomic material was not expected under the conditions used for nuclear extract preparation (see *Supplementary Methods*, section *Subcellular fractionation and HSV-1 genomic qPCR*). Indeed, qPCR analysis of the second wash fraction before nuclear lysis revealed near-complete absence of DNA derived from the incoming virus.

REVIEWER #2, COMMENT 4:

4.1) Figure 5. The pGL-T9G reporter system, originally using a stably transfected ORF9 luciferase reporter MeWo cell line, is elegant and has been validated for quantifying VZV infection and HSV-1 infection but I have not seen this system based on transient transfection of the reporter in HeLa cells.

Has this been validated to show strict correlation between HSV infection and luciferase reporter expression?

Using the pGL-T9G reporter system, we now assessed the relationship between luciferase reporter expression by transient transfection and HSV-1 infection. Indeed, the virus-induced reporter signal augmented with time and increasing MOI and could be antagonised by addition of the HSV-specific DNA polymerase inhibitor phosphonoacetic acid (PAA²⁰, see Supplementary Fig. 6c-e). Moreover, expression of the immediate-early protein VP22 showed a strong correlation with our reporter (see Supplementary Fig. 6c, lower panel). Together, these results clearly demonstrate that the reporter activity was entirely dependent on HSV-1 infection.

4.2) It would be much better to recapitulate the effect or lack of effect of these MxB mutants by stable transfection in e.g. A459 cells followed by superinfection with HSV-1.

We initially evaluated the possibility of generating stably transduced A549 cells with our different MxB mutant vectors. However, stable expression of MxB variants requires sub-cloning of transduced cell populations, since in the case of A549 cells, such populations exhibited MxB protein expression in only about 5-10% of all cells (see also our response to Reviewer #1, comment 2). Thus, we decided to use the transient expression system that allowed us to study the behaviour of different MxB variants in a dose-dependent manner. The transient expression system used has the additional advantage that it measures virus-induced transcription only in transfected cells, *i.e.* cells that likely harbour both the luciferase reporter vector and the overexpression vector.

References

1. Goujon, C. *et al.* Human MX2 is an interferon-induced post-entry inhibitor of HIV-1 infection. *Nature* **502**, 559-562 (2013).
2. Wulff, N. H., Tzatzaris, M. & Young, P. J. Monte Carlo simulation of the Spearman-Kaerber TCID50. *Journal of clinical bioinformatics* **2**, 5 (2012).
3. Schneider, W. M., Chevillotte, M. D. & Rice, C. M. Interferon-stimulated genes: a complex web of host defenses. *Annual review of immunology* **32**, 513-545 (2014).
4. Al-khatib, K., Williams, B. R., Silverman, R. H., Halford, W. & Carr, D. J. The murine double-stranded RNA-dependent protein kinase PKR and the murine 2',5'-oligoadenylate synthetase-dependent RNase L are required for IFN-beta-mediated resistance against herpes simplex virus type 1 in primary trigeminal ganglion culture. *Virology* **313**, 126-135 (2003).
5. Carr, D. J., Tomanek, L., Silverman, R. H., Campbell, I. L. & Williams, B. R. RNA-dependent protein kinase is required for alpha-1 interferon transgene-induced resistance to genital herpes simplex virus type 2. *Journal of virology* **79**, 9341-9345 (2005).
6. Duus, K. M., Lentchitsky, V., Wagenaar, T., Grose, C. & Webster-Cyriaque, J. Wild-type Kaposi's sarcoma-associated herpesvirus isolated from the oropharynx of immune-competent individuals has tropism for cultured oral epithelial cells. *Journal of virology* **78**, 4074-4084 (2004).
7. Cook, R. D. *et al.* Tracking familial transmission of Kaposi's sarcoma-associated herpesvirus using restriction fragment length polymorphism analysis of latent nuclear antigen. *Journal of virological methods* **105**, 297-303 (2002).
8. Cook, R. D. *et al.* Mixed patterns of transmission of human herpesvirus-8 (Kaposi's sarcoma-associated herpesvirus) in Malawian families. *The Journal of general virology* **83**, 1613-1619 (2002).
9. Boshoff, C. *et al.* Kaposi's sarcoma-associated herpesvirus infects endothelial and spindle cells. *Nature medicine* **1**, 1274-1278 (1995).
10. Cai, Q., Verma, S. C., Lu, J. & Robertson, E. S. Molecular biology of Kaposi's sarcoma-associated herpesvirus and related oncogenesis. *Advances in virus research* **78**, 87-142 (2010).
11. Horvath, C. M., Stark, G. R., Kerr, I. M. & Darnell, J. E., Jr. Interactions between STAT and non-STAT proteins in the interferon-stimulated gene factor 3 transcription complex. *Molecular and cellular biology* **16**, 6957-6964 (1996).
12. Vieira, J. & O'Hearn, P. M. Use of the red fluorescent protein as a marker of Kaposi's sarcoma-associated herpesvirus lytic gene expression. *Virology* **325**, 225-240 (2004).
13. Jeong, J. H., Hines-Boykin, R., Ash, J. D. & Dittmer, D. P. Tissue specificity of the Kaposi's sarcoma-associated herpesvirus latent nuclear antigen (LANA/orf73) promoter in transgenic mice. *Journal of virology* **76**, 11024-11032 (2002).
14. Wang, I. H. *et al.* Tracking viral genomes in host cells at single-molecule resolution. *Cell host & microbe* **14**, 468-480 (2013).
15. Flatt, J. W. & Greber, U. F. Misdelivery at the Nuclear Pore Complex-Stopping a Virus Dead in Its Tracks. *Cells* **4**, 277-296 (2015).
16. Sekine, E., Schmidt, N., Gaboriau, D. & O'Hare, P. Spatiotemporal dynamics of HSV genome nuclear entry and compaction state transitions using bioorthogonal chemistry and super-resolution microscopy. *PLoS pathogens* **13**, e1006721 (2017).
17. Rode, K. *et al.* Uncoupling uncoating of herpes simplex virus genomes from their nuclear import and gene expression. *Journal of virology* **85**, 4271-4283 (2011).
18. Sodeik, B., Ebersold, M. W. & Helenius, A. Microtubule-mediated transport of incoming herpes simplex virus 1 capsids to the nucleus. *The Journal of cell biology* **136**, 1007-1021 (1997).
19. Timm, A. & Yin, J. Kinetics of virus production from single cells. *Virology* **424**, 11-17 (2012).
20. Overby, L. R. *et al.* Inhibition of herpes simplex virus replication by phosphonoacetic acid. *Antimicrobial agents and chemotherapy* **6**, 360-365 (1974).

REVIEWERS' COMMENTS:

Reviewer #2 (Remarks to the Author):

The manuscript by Crameri et al is a revised version of a manuscript I originally reviewed which this reviewer believed required a number of additional analyses and clarifications to more fully support the conclusions of the authors.

It has to be said that the authors have made substantial efforts to address these comments/queries which, I believe, now makes the manuscripts conclusions far more robust.

A minor comment, which need to be addressed is as follows:

(i) Whilst I understand that the authors want to make their data as impactful as possible, it gives a false impression to state that knock-down of MX2 resulted in a 7.5-25 fold increase of viral titres in the presence of interferon (line 96) – the statistically relevant data show that this is no more than

10-fold on average and it should be stated as such. This roughly “10-fold” effect is actually supported by many of the comparative data in other figures.

Response to Reviewers:

Reviewers comment: The manuscript by Crameri et al is a revised version of a manuscript I originally reviewed which this reviewer believed required a number of additional analyses and clarifications to more fully support the conclusions of the authors.

It has to be said that the authors have made substantial efforts to address these comments/queries which, I believe, now makes the manuscripts conclusions far more robust.

A minor comment, which need to be addressed is as follows:

(i) Whilst I understand that the authors want to make their data as impactful as possible, it gives a false impression to state that knock-down of MX2 resulted in a 7.5-25 fold increase of viral titres in the presence of interferon (line 96) – the statistically relevant data show that this is no more than 10-fold on average and it should be stated as such. This roughly “10-fold” effect is actually supported by many of the comparative data in other figures

Response to the reviewers comment. We agree with the reviewer. We have changed our statement according to the reviewer’s suggestion and state now:

Line 96: Remarkably, in T98G cells transfected with siRNAs specific for *MX2*, the observed IFN-mediated inhibition of HSV-1 infection was partially released, resulting in **approximately 10-fold increased viral titres**.

We would like to thank the reviewers for their fair and constructive reviews.